# Scaling Multimodal Temporal Graphs with Event-Adaptive Compression and Sparse Connectivity

## Abstract

Multimodal temporal data analysis presents a challenge: it needs to strike a balance between high resolution for capturing sudden events and a wide temporal range for scalability. This often results in vast graph models that can be computationally intractable. Current approaches tend to either break the sequences into fixed-length segments or trim edges to stay within budget constraints, often at the cost of fidelity.

We introduce **EAMC–C2SG**, a novel framework that dynamically compresses temporal streams into segments tailored to events and creates a sparse graph model that respects temporal ordering. By curbing the proliferation of nodes and edges, our design achieves strict budget control while reducing complexity from a quadratic to a near-linear scale with respect to sequence length.

Our framework preserves valuable information in multimodal temporal data and, when tested on extensive clinical datasets (MIMIC-IV + CXR) and diverse cross-domain benchmarks (TimeMMD), achieves superior performance on the evaluated tasks with markedly lower latency and memory usage. Beyond raw performance, EAMC–C2SG also offers interpretable segmentations and insightful graph diagnostics, making it a scalable and transparent solution for multimodal temporal learning.

## 1 Introduction

Multimodal temporal data are ubiquitous in real-world applications, ranging from healthcare monitoring with physiological signals and medical notes, to video understanding that integrates visual frames, audio, and text transcripts. A central challenge lies in the coexistence of heterogeneous sampling rates and multi-scale temporal dependencies: fine-grained resolution is necessary to capture abrupt local events, whereas coarse-grained resolution better suits long-term dynamics. Designing a representation that adapts across these scales while remaining computationally tractable remains an open problem.

Representing multimodal sequences as graphs provides a natural way to unify heterogeneous modalities within a relational space, enabling structured reasoning and interpretability that are particularly valuable in domains such as healthcare (Rossi et al., 2020; Liu et al., 2024a). However, this flexibility comes at the cost of what we term the "node explosion" problem: each modality contributes a large number of nodes, and edge counts grow rapidly with sequence length, leading to quadratic complexity in message passing. Graph neural networks (GNNs), though powerful for relational reasoning, thus become prohibitively expensive in both memory and computation when applied to long multimodal streams.

Existing approaches have sought to mitigate this tension but fall short in key respects. Fixed-window or pyramid-based multiscale representations (Liu et al., 2022; Wu et al., 2023) often over-tokenize stable regions by applying uniform resolution regardless of local dynamics and suffer from boundary-cutting artifacts that fragment events. Fully connected attention or correlation graphs (Zaheer et al., 2020; Zhang et al., 2020) introduce excessive edges, leading to quadratic complexity in message passing. Pooling or random sampling strategies reduce complexity but frequently sacrifice

interpretability and accuracy (Bolya et al., 2023). As a result, none of these methods simultaneously achieve compactness, temporal adaptivity, and scalability.

In this work, we propose **Event-Adaptive Multiscale Compression with Sparse Graphs (EAMC–C2SG)**, a framework that reconciles fidelity and scalability by jointly addressing temporal segmentation and graph construction. First, the event-adaptive segmentation (EAMC) module allocates fine resolution to abrupt, high-entropy regions while compressing stable intervals, thereby maximizing information retention and reducing redundancy. Second, the node budget controller (NBC) enforces a global constraint on the number of retained segments, merging unselected ones into nearby nodes to preserve local continuity. This budget-aware compression prevents node explosion, effectively turning the redundancy common in low information-density or noisy clinical data into efficiency. Third, sparse graphs are constructed over these super-nodes with strict ordering and bounded neighborhoods, ensuring leakage-free message passing at near-linear complexity.

Our contributions are summarized as follows:

- We formulate the tension between multiscale heterogeneity and graph scalability as a central bottleneck in multimodal temporal learning.
- We propose EAMC–C2SG, a unified framework that integrates event-adaptive compression, budget-constrained selection, and sparse connectivity.
- We demonstrate significant efficiency gains with minimal accuracy loss, consistently outperforming strong baselines on multimodal temporal benchmarks including MIMIC-IV + CXR and TimeMMD.
- We provide interpretability analyses through event-level segmentation, sparsity diagnostics, and modality-level attribution, highlighting the transparency of the learned representations.

Taken together, these contributions fill a missing intersection in the literature: existing methods either rely on static windows or dense graphs, but none simultaneously achieve adaptive resolution, compact and interpretable graph construction, and budget-aware scalability. Our framework explicitly addresses this gap, offering a scalable solution that is especially suited for domains with noisy signals and redundant measurements. Extensive experiments show that EAMC–C2SG achieves superior performance on the evaluated tasks while using only a fraction of the nodes and memory. As illustrated in Figure 1, it shifts the Pareto frontier of efficiency–accuracy trade-offs and preserves high fidelity across a wide range of node budgets, demonstrating its ability to reconcile scalability with predictive performance.

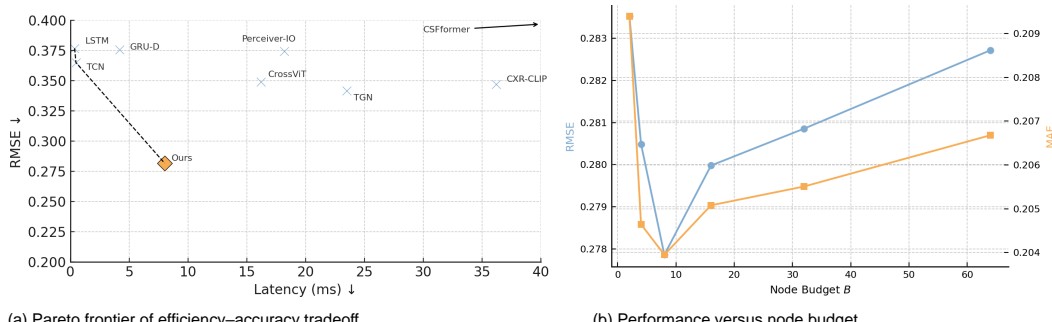

(a) Pareto frontier of efficiency–accuracy tradeoff.  (b) Performance versus node budget

Figure 1: **Efficiency–accuracy tradeoff and scalability.** **(a)** Pareto frontier of RMSE vs. latency, where our method achieves lower error with tractable runtime compared to baselines. **(b)** RMSE (blue) and MAE (orange) across node budget $B$, showing stable fidelity under strict budgets.

## 2 RELATED WORK

### 2.1 MULTIMODAL TEMPORAL MODELING

Multimodal temporal data are often irregular and asynchronous. Continuous-time neural models such as Neural ODEs, Neural CDEs, latent trajectory methods, and GRU-ODE-Bayes handle irregular sampling through continuous dynamics, while neural Hawkes processes and temporal point

processes capture event-level patterns (Chen et al., 2018; Rubanova et al., 2019; Kidger et al., 2020; De Brouwer et al., 2019; Mei & Eisner, 2017). Perceiver and Perceiver IO unify heterogeneous inputs via cross-attention bottlenecks (Jaegle et al., 2021; 2022), and large vision–language models such as CLIP, Flamingo, and BLIP-2 deliver strong cross-modal alignment (Radford et al., 2021; Alayrac et al., 2022; Li et al., 2023). However, these approaches mainly target representation alignment and do not fully resolve scale mismatch or causal validity in long multimodal streams.

## 2.2 SCALING AND SPARSIFICATION FOR LONG SEQUENCES

Scaling to long sequences has driven efficient Transformer designs based on sparse patterns, low-rank projections, or kernelization (Reformer, Linformer, BigBird, Longformer, Performer, Nyströmformer) and recent implementation-level improvements such as FlashAttention (Kitaev et al., 2020; Wang et al., 2020; Zaheer et al., 2020; Beltagy et al., 2020; Choromanski et al., 2021; Dao et al., 2022; Dao, 2023; Shah et al., 2024). Time-series Transformers introduce trend or spectral priors, including Informer, Autoformer, Pyraformer, FEDformer, Crossformer, PatchTST, Times-Net, and iTransformer (Zhou et al., 2021; Wu et al., 2021; Liu et al., 2022; Zhou et al., 2022; Zhang & Yan, 2023; Nie et al., 2023; Wu et al., 2023; Liu et al., 2024b). Yet most rely on static windows or pyramids (Zhou et al., 2021; Nie et al., 2023; Wu et al., 2023), which over-tokenize stable periods (e.g., using fixed-size patches in PatchTST) and cut through event boundaries, creating quadratic growth in node and edge counts once cast as graphs.

Dynamic sparsification methods such as TokenLearner, DynamicViT, EViT, and ToMe adaptively prune or merge tokens (Ryoo et al., 2021; Rao et al., 2021; Liang et al., 2022; Bolya et al., 2023), and mixture-of-experts models such as Switch Transformer and GLaM leverage sparse activation (Fedus et al., 2021; Du et al., 2022). TimeFilter-style filtering (Hu et al., 2025) applies fixed patches followed by selection but risks boundary artifacts and temporal leakage. These approaches rarely enforce causal constraints or provide explicit node/edge budgets for downstream graphs.

## 2.3 GRAPHS, CAUSALITY, AND ADAPTIVE COMPUTATION

Graph pooling and coarsening methods (DiffPool, TopKPool, SAGPool, ASAP, MinCutPool, Graph U-Net) compress nodes while retaining task signals (Ying et al., 2018; Gao & Ji, 2019; Lee et al., 2019; Ranjan et al., 2020; Bianchi et al., 2020). Temporal and dynamic GNNs (DyRep, TGAT, TGN) model evolving neighborhoods with time encodings or memory (Trivedi et al., 2019; Xu et al., 2020; Rossi et al., 2020), but they scale poorly for multimodal graphs and seldom guarantee temporal causality. Classical Granger causality and modern conditional-independence approaches highlight the need for directed lagged edges and explicit windows (Granger, 1969; Runge et al., 2019; Runge, 2020), while recent analyses emphasize the pitfalls of positive lags (Jhin et al., 2024). Adaptive computation methods, such as ACT, BranchyNet, and Gumbel-Softmax, align compute with input difficulty (Teerapittayanon et al., 2016; Jang et al., 2017; Maddison et al., 2017), and unlike post-hoc pruning, budget-aware training jointly shapes representations and graph topology under explicit complexity constraints.

Taken together, these threads highlight the need for an integrated approach that combines event-adaptive segmentation, causal graph construction, and budget-aware sparsification. A more detailed discussion and extended analysis of related work are provided in Appendix D.

## 3 METHOD

For clarity, we summarize all key symbols before describing each module; Appendix B defines the unified notation used throughout Section 3.

**Overview.** Our framework contains three sequential modules summarized in Table 1. Together, they compress long temporal streams, enforce strict node budgets, and construct causally valid sparse graphs for multimodal reasoning.

Figure 2 provides an overview of the full workflow. We first outline the shared token space, then detail each module and the resulting training objective.

Table 1: Overview of the EAMC–C2SG framework.

| Module | Function and Role |
|--------|-------------------|
| **EAMC** | Event-Adaptive Multiscale Compression: partitions a long temporal sequence into event-aware segments and derives high-fidelity segment embeddings. |
| **NBC** | Node Budget Controller: selects or merges segments under a fixed node budget $B$, ensuring a compact and scalable graph. |
| **C2SG** | Causal-Constrained Sparse Graph: builds a directed, lag-bounded graph with top-$\kappa$ sparsification for multimodal fusion while preserving temporal causality. |

Figure 2: **Overall architecture of EAMC–C2SG.** Multimodal patient data are mapped into a shared token space, where EAMC forms segment-level nodes with IRR/BCR diagnostics, NBC enforces a fixed node budget, and C2SG applies causal masks and sparse cross-modal edges for efficient and interpretable prediction.

We introduce **EAMC–C2SG** as a budget-aware pipeline operating entirely in a shared token space of width $d$. The design principle is straightforward: assign higher temporal resolution to regions with events, compress stable regions into *super nodes*, and restrict message passing to causally valid edges under explicit compute budgets.

Concretely, modality-specific extractors produce shared-space tokens for images, text, a static vector, and an irregular time-series stream. EAMC computes boundary scores $\mathcal{B}$, aggregates each segment into a representation with IRR/BCR diagnostics, and produces super nodes. NBC then retains at most $B$ segments via an index set $\mathcal{I}_B$, optionally merging nearby discarded segments to preserve local evidence without increasing node count.

Over the ordered and budgeted nodes, C2SG applies a lag-window mask of size $\epsilon$ and a per-node edge limit $\kappa$, forming a directed sparse graph that scales near-linearly in $B$. A lightweight GNN produces a pooled summary $\mathbf{g}$, which is fused with image/text/static tokens through temperature-scaled cross-modal attention to form the final representation $\mathbf{f}$. Training uses a unified objective balancing task performance with information retention and boundary quality.

The remaining subsections describe EAMC (event-salient segmentation), NBC (budgeted selection and merging), C2SG (causal sparse graph construction), and the cross-modal fusion procedure with the full training protocol.

### 3.1 FEATURE EXTRACTORS AND SHARED SPACE

All modalities are first projected into a shared $d$-dimensional token space through modality-specific encoders followed by linear projection. In particular, we denote the time-series token matrix as $\mathbf{Z} \in \mathbb{R}^{T \times d}$ with rows $\mathbf{z}_t$, and its mean $\bar{\mathbf{z}}$ is used in diagnostic metrics (e.g., IRR$^+$). Image tokens $\mathbf{Z}^{\text{img}}$, text tokens $\mathbf{Z}^{\text{txt}}$, and a static feature vector $\mathbf{z}^{\text{sta}}$ are defined analogously, but for clarity we defer their detailed notation to Appendix C. All subsequent modules—EAMC, NBC, C$^2$SG, fusion, and task heads—operate exclusively on these shared-space representations.

### 3.2 EVENT-ADAPTIVE MULTISCALE COMPRESSION (EAMC)

Event-Adaptive Multiscale Compression (EAMC) transforms a time-series token stream $\mathbf{Z} \in \mathbb{R}^{T \times d}$ into a compact set of high-fidelity super-nodes that preserve event salience while suppressing redundancy in stable regions. This process yields three outputs: (i) a sequence of learned boundaries $\mathcal{B} = \{b_0, \ldots, b_S\}$, (ii) segment-specific token matrices $\mathbf{Z}_s \in \mathbb{R}^{L_s \times d}$, and (iii) segment-level representations $\mathcal{M} = \{\mathbf{m}_s\}_{s=1}^{S}$, which are passed to the downstream NBC for selection and routing.

The module receives as input the row-major time-series tokens $\mathbf{z}_t$ (rows of $\mathbf{Z}$), and produces:

- a boundary sequence $\mathcal{B}$ such that $0 = b_0 < b_1 < \cdots < b_S = T$,
- a segment-wise slice $\mathbf{Z}_s = \mathbf{Z}[b_{s-1}{:}b_s, :]$,
- a single feature vector $\mathbf{m}_s \in \mathbb{R}^d$ for each segment,
- diagnostic signals for assessing information retention and boundary integrity.

To identify meaningful transitions, we directly monitor the change rate of segmentation probabilities derived from attention outputs. Each token is first enhanced by a multi-head self-attention block, and a lightweight MLP maps it to a segmentation probability $P(t) \in [0, 1]$:

$$P(t) = \sigma(\text{MLP}(\text{Attn}(z_t))). \tag{1}$$

We then compute the absolute change rate between consecutive steps, $\Delta P(t) = |P(t) - P(t-1)|$, apply a short moving average to suppress noise, and declare candidate boundaries when

$$\text{Boundary}(t) = \mathbf{1}\{\Delta P(t) > \theta\}, \tag{2}$$

where $\theta$ is a predefined threshold. To avoid trivial fragmentation, we impose a minimum segment length and limit the maximum number of segments by selecting the top-$k$ peaks of $\Delta P(t)$.

Within each segment, the goal is to capture both abrupt local transients and longer oscillatory patterns. To this end, EAMC applies a gated multiscale time–frequency fusion (GMA) mechanism: a temporal branch aggregates multi-scale structure with a bank of convolutions followed by attention and pooling to produce $\overline{\mathbf{u}}_s \in \mathbb{R}^d$, while a complementary frequency branch, implemented with Fourier or wavelet transforms, yields $\overline{\mathbf{v}}_s \in \mathbb{R}^d$. A learnable gate balances their contributions, ensuring that both short-term events and long-term rhythms are preserved:

$$m_s = g_{tf,s}\, u_s + (1 - g_{tf,s})v_s, \qquad g_{tf,s} = \sigma(W_g[u_s; v_s] + b_g). \tag{3}$$

To facilitate downstream selection, each segment is assigned an importance score $\alpha_s = \sigma(\text{MLP}(m_s))$. During inference, the Node Budget Controller (NBC) selects a subset $\mathcal{I}_B$ of $\hat{S} \leq B$ segments according to these scores. For unselected segments, EAMC optionally propagates their evidence to nearby retained nodes with temporally decaying weights $\gamma_{is} \propto \exp(-|c_i - c_s|/\tau_m)$, forming merged vectors

$$\tilde{\mathbf{m}}_i = \frac{\sum_{s \in \mathcal{N}(i)} \gamma_{is}\mathbf{m}_s}{\sum_{s \in \mathcal{N}(i)} \gamma_{is}}. \tag{4}$$

Finally, EAMC computes two diagnostic metrics to evaluate compression quality. Information retention is quantified via a lightweight decoder that reconstructs $\tilde{\mathbf{Z}} = \text{Dec}(\mathcal{M}, \mathcal{B})$, with reconstruction loss

$$L_{\text{ae}} = \frac{1}{T}\sum_{t=1}^{T} \|z_t - \tilde{z}_t\|_2^2, \qquad \sigma^2 = \frac{1}{T}\sum_{t=1}^{T} \|z_t - \bar{z}\|_2^2, \tag{5}$$

where $\bar{z}$ denotes the sequence mean. The Information Retention Rate (IRR) is then

$$\text{IRR}^+ = \max\left\{0,\, 1 - \tfrac{L_{ae}}{\sigma^2}\right\}, \tag{6}$$

where we adopt the truncated non-negative form, with "IRR" in tables and figures denoting $\text{IRR}^+$. The Boundary Cutting Rate (BCR) measures alignment between event windows and segment boundaries. Using the soft assignment matrix $\mathbf{S}^{\text{seg}} \in [0,1]^{T \times S}$, the soft coverage of an event window $\mathcal{W}_e$ is

$$q_{e,s} = \frac{1}{|\mathcal{W}_e|} \sum_{t \in \mathcal{W}_e} S^{\text{seg}}[t,s], \qquad \sum_{s=1}^{S} q_{e,s} = 1, \tag{7}$$

and the overall cutting rate is

$$\text{BCR}_{\text{soft}} = \frac{1}{|\mathbb{E}|} \sum_{e \in \mathbb{E}} \left(1 - \sum_{s=1}^{S} q_{e,s}^2\right). \tag{8}$$

Low values of $\text{BCR}_{\text{soft}}$ indicate that events are well captured within intact segments rather than being fragmented across many, ensuring that the diagnostic remains meaningful across both annotated and unannotated datasets.

## 3.3 NODE BUDGET CONTROLLER (NBC)

The Node Budget Controller (NBC) enforces a strict upper bound on the number of segment nodes retained for downstream processing. Operating after dynamic segmentation, NBC receives the set of segment-level representations $\mathbf{M} \in \mathbb{R}^{B \times S \times d}$ together with their importance scores $\boldsymbol{\alpha} \in \mathbb{R}^{B \times S}$, and compresses them into at most $B$ nodes. The goal is to preserve the most informative segments while ensuring that the token budget is not exceeded.

NBC performs a top-$B$ selection according to the importance scores, yielding a preserved index set $I_B$ with $\hat{S} = |I_B| \leq B$. The remaining segments are merged into their nearest preserved neighbors along the temporal axis, thereby maintaining local continuity. Formally, the assignment of each segment $s$ (for sample $b$) is defined as

$$\text{assign}[b,s] = \begin{cases} \text{index of } s & \text{if } s \in I_B, \\ \arg\min_{i \in I_B} |i - s| & \text{otherwise.} \end{cases} \tag{9}$$

From this assignment, NBC constructs a one-hot assignment tensor $\Pi \in \{0,1\}^{B \times S \times \hat{S}}$ with

$$\Pi[b,s,k] = \begin{cases} 1 & \text{if assign}[b,s] = k, \\ 0 & \text{otherwise.} \end{cases} \tag{10}$$

Each merged node is then computed by locality-aware averaging:

$$\tilde{\mathbf{M}}[b,k] = \frac{\sum_{s=1}^{S} \Pi[b,s,k]\, \mathbf{M}[b,s]}{\max\left(1, \sum_{s=1}^{S} \Pi[b,s,k]\right)}. \tag{11}$$

This mechanism combines hard selection with deterministic local merging. The top-$B$ step guarantees a fixed budget, while the averaging step ensures that unselected segments are absorbed without loss of temporal coverage. As a result, NBC provides a scalable and differentiable compression layer that preserves event fidelity while significantly reducing computational cost. It is particularly suited to long-sequence modeling and real-time inference scenarios, where efficiency is critical.

## 3.4 CAUSAL-CONSTRAINED SPARSE GRAPH

The Causal-Constrained Sparse Graph (C2SG) module enforces *structural causality*—temporal ordering with a bounded lag—while enabling *multimodal fusion* within a unified graph representation. Its objective is to guarantee leakage-free message passing across temporal nodes, while selectively

integrating complementary information from images and text. The resulting backbone is both computationally efficient and structurally interpretable.

Formally, let $\mathbf{H}^{\text{ts}} \in \mathbb{R}^{B \times S \times d}$ denote temporal segment representations, with optional image and text embeddings $\mathbf{H}^{\text{img}}$ and $\mathbf{H}^{\text{txt}}$. The heterogeneous node set is defined as

$$\mathbf{H} = \{\mathbf{H}^{\text{ts}}, \mathbf{H}^{\text{img}}, \mathbf{H}^{\text{txt}}\}. \tag{12}$$

Temporal nodes are arranged according to sequence order, while image and text tokens enter as auxiliary vertices. To ensure causal consistency among temporal nodes, a binary mask $\mathbf{M}^{\text{ts}} \in \{0,1\}^{B \times S \times S}$ is constructed such that only past nodes within a lag window $\epsilon$ are permitted as predecessors:

$$M[b,i,j] = \mathbf{1}(j < i \,\wedge\, |i-j| \le \epsilon). \tag{13}$$

The mask is normalized row-wise to yield $\tilde{\mathbf{M}}^{\text{ts}}$, which serves as a valid distribution over causal neighbors.

For cross-modal relations, no temporal ordering is imposed; candidate edges between temporal and auxiliary nodes are instead scored by cosine similarity and gated through a sigmoid function. Importantly, all auxiliary modalities attach only to the *current* temporal node, and never create edges to earlier temporal positions. Thus, cross-modal connections cannot form future→past paths nor bypass the temporal mask imposed on the backbone. The unified edge score is therefore defined as

$$w[b,i,j] = \sigma\left(\frac{h_{b,i}^{\top} h_{b,j}}{\|h_{b,i}\| \, \|h_{b,j}\|}\right) \cdot M[b,i,j], \tag{14}$$

where $M[b,i,j]$ denotes the causal mask for temporal pairs and takes the value 1 for cross-modal pairs. To control sparsity, each node retains only its top-$\kappa$ inbound edges, yielding a multimodal adjacency that combines temporally causal structure with selectively integrated visual and textual evidence. This formulation aligns with the general principle of multimodal graph neural networks Zhang et al. (2020); Peng et al. (2021), where heterogeneous modalities are fused through graph-based message passing.

Message passing is then performed on the sparse adjacency using $L$ residual GNN layers:

$$h^{(\ell)} = h^{(\ell-1)} + \text{Dropout}\Big(\text{GELU}\big(W^{(\ell)}(\tilde{M} h^{(\ell-1)}) + b^{(\ell)}\big)\Big), \qquad \ell = 1, \dots, L. \tag{15}$$

The complexity of this construction is near-linear in the node budget $B$, since temporal neighborhoods are bounded by $O(S \cdot \epsilon)$ and multimodal edges are restricted to top-$\kappa$ connections. To enhance interpretability, we further report summary statistics of the constructed graphs, including the number of nodes $|V|$, edges $|E|$, average degree $\bar{d}$, and estimated latency. Through this design, C2SG yields an efficient and interpretable multimodal causal graph, strictly preventing future-to-past leakage while enabling external modalities to enrich temporal reasoning.

We train the model with a composite objective consisting of a task-specific prediction loss, a reconstruction loss from the EAMC autoencoder, a cross-modal contrastive alignment loss, and an auxiliary binary cross-entropy loss for missingness prediction:

$$\mathcal{L}_{\text{total}} = \mathcal{L}_{\text{task}} + \lambda_{\text{rec}}\mathcal{L}_{\text{rec}} + \lambda_{\text{cl}}\mathcal{L}_{\text{cl}} + \lambda_{\text{miss}}\mathcal{L}_{\text{BCE}}, \tag{16}$$

where $(\lambda_{\text{rec}}, \lambda_{\text{cl}}, \lambda_{\text{miss}}) = (1.0, 0.1, 0.01)$ unless otherwise specified. This version is the definitive training objective; the appendix version will be updated to match and retain only additional diagnostic penalty terms for constraint analysis.

Unless otherwise stated, the causal lag window is set to $\epsilon = 4$, and each node is allowed at most $\kappa = 2$ incoming edges. These values are chosen to balance temporal coverage and sparsity.

For cross-modal alignment, we adopt a CLIP-style scheme with bidirectional InfoNCE over all pairwise modality combinations. Fusion across modalities is performed via multi-head self-attention with temperature scaling. Training uses cosine annealing with warm-up, mixed precision, gradient clipping, and the Muon optimizer, with early stopping on validation performance. All experiments are run on NVIDIA 4090 GPUs.

# 4 EXPERIMENTS

We conduct extensive experiments to evaluate the effectiveness, efficiency, and generality of our proposed method. Our evaluation focuses on (i) clinical prediction with rich multimodal data from MIMIC-IV and MIMIC-CXR, and (ii) cross-domain generalization on the TimeMMD benchmark, which covers diverse multimodal temporal domains. We also provide ablation studies, graph-level diagnostics, budget–performance tradeoffs, and sparsity analysis.

## 4.1 EXPERIMENTAL SETUP

**Datasets.** We evaluate on two multimodal benchmarks: (i) a patient-aligned cohort from MIMIC-IV and MIMIC-CXR, integrating chest X-rays, reports, physiological time series, and demographics, serving as a realistic clinical risk prediction benchmark; (ii) TimeMMD, covering nine domains of multimodal temporal sequences with paired text, smaller in scale but complete without missing values, enabling controlled cross-domain analysis. Dataset preprocessing and statistics are detailed in Appendix C.

**Tasks, Metrics, and Implementation.** On MIMIC, we perform next-step prediction over 20 physiological variables, using MAE, MSE, and RMSE for evaluation; on TimeMMD, we follow its forecasting protocol and report MSE and MAE. Efficiency is measured by latency (ms/sample), and diagnostics include IRR, BCR, and graph statistics ($|V|, |E|, \bar{d}$). All models are trained with the Muon optimizer (weight decay $10^{-5}$, initial LR $3 \times 10^{-4}$) under cosine annealing with warm-up. To reduce cost, ResNet-50 and BERT backbones are frozen while training only fusion, segmentation, graph, and prediction modules, with mixed-precision enabled throughout.

**Baselines.** We compare against representative methods from three categories: (i) clinical time-series models, including *GRU-D*, *LSTM*, *TCN*, and three general-purpose state-of-the-art forecasting architectures—*TimesNet* (Wu et al., 2023), *iTransformer* (Liu et al., 2024b), and *PatchTST* (Nie et al., 2023); (ii) multimodal or static fusion models, including *CSFformer* (Wan et al., 2025), *CXR-CLIP* (Radford et al., 2021), and *CrossViT* (Chen et al., 2021); (iii) general latent or graph architectures, including *Perceiver-IO* (Jaegle et al., 2022) and *TGN* (Rossi et al., 2020).

All baselines use the same embedding width $d$, optimizer, training schedule, and a capped node budget $B$. As shown in Table 2, our method achieves consistently lower regression error with competitive efficiency.

Table 2: Main results on MIMIC-IV + CXR. Baselines are grouped as: **Time-Series (TS) only** (GRU-D, LSTM, TCN, iTransformer, PatchTST, TimesNet); **Multimodal (MM)** (CSFformer, CXR-CLIP, CrossViT); **General-purpose** (TGN, Perceiver-IO). Lower values indicate better performance.

| Metric | TS-only | | | | | | Multimodal | | | General | | |
| | GRU-D | LSTM | TCN | iTransformer | PatchTST | TimesNet | CSFf. | CXR-CLIP | CrossViT | TGN | Per-IO | **Ours** |
|---|---|---|---|---|---|---|---|---|---|---|---|---|
| MSE | 0.1411 | 0.1420 | 0.1332 | 0.0992 | 0.0891 | 0.0927 | 0.1575 | 0.1204 | 0.1218 | 0.1203 | 0.1400 | **0.0793** |
| RMSE | 0.3756 | 0.3768 | 0.3650 | 0.2969 | 0.2902 | 0.3018 | 0.3968 | 0.3470 | 0.3489 | 0.3418 | 0.3742 | **0.2818** |
| MAE | 0.2693 | 0.2698 | 0.2687 | 0.2290 | 0.2274 | 0.2327 | 0.3079 | 0.2488 | 0.2399 | 0.2339 | 0.2884 | **0.2060** |
| Latency (ms) | 4.18 | **0.36** | 0.47 | 2.017 | 2.736 | 148.732 | 363.9 | 36.2 | 16.2 | 23.5 | 18.2 | 8.01 |

Table 3: Ablation on MIMIC-IV + CXR. Metrics: graph size $(V, E)$, diagnostics (IRR ($\mathrm{IRR}^+$), BCR), efficiency (latency), and errors (MSE, RMSE). Higher is better for IRR ($\mathrm{IRR}^+$); lower for others. Best in **bold**.

| **Variant** | $V$ | $E$ | $\mathrm{IRR}^+$ | **BCR** $\downarrow$ | Latency(ms) | **MSE** $\downarrow$ | **RMSE** $\downarrow$ |
|---|---|---|---|---|---|---|---|
| Full model | 16 | 36 | 0.9903 | 0.0422 | 9.45 | **0.0793** | **0.2818** |
| --Multimodal | 16 | 36 | 0.9903 | 0.0422 | **4.69** | 0.0903 | 0.3170 |
| --Causal | 16 | 256 | 0.9903 | 0.0422 | 12.32 | 0.0794 | 0.2822 |
| --DynSeg | 16 | 36 | 0.8211 | 0.2397 | 8.24 | 0.0815 | 0.2827 |
| --NBC | 64 | 180 | 0.9903 | 0.0422 | 10.30 | 0.0799 | 0.2827 |

## 4.2 ABLATION AND MODULE DIAGNOSIS

We evaluate the role of each module through ablations on **MIMIC-IV + CXR**, by disabling: `--Multimodal` (replace image and text features with zero vectors while keeping dimensionality), `--DynSeg` (uniform segmentation of time series without event adaptivity), `--Causal` (remove the causal lag mask in the GNN), and `--NBC` (disable the node budget controller, passing all segmented nodes directly into the GNN). We report predictive accuracy (MSE, RMSE), efficiency (latency), diagnostic measures (IRR, BCR), and graph statistics ($V$, $E$). Results are shown in Table 3.

The complete EAMC–C2SG delivers the best overall balance across metrics. Removing causal masking (`--Causal`) achieves a lower MSE but violates temporal validity by allowing positive-lag leakage. Eliminating multimodal fusion (`--Multimodal`) substantially worsens RMSE, showing that text and image provide complementary signals to time-series dynamics. Skipping event-adaptive segmentation (`--DynSeg`) reduces IRR and increases BCR, indicating poor boundary alignment and degraded fidelity. Finally, disabling the budget controller (`--NBC`) inflates graph size and latency, undermining efficiency. Overall, these results confirm that each component makes a substantive contribution, and only their integration enables accurate, efficient, and scalable multimodal temporal graph learning.

## 4.3 CROSS-DOMAIN GENERALIZATION ON TIMEMMD

To assess the generality of our method beyond clinical data, we evaluate on the TimeMMD benchmark. We focus on the **Energy** domain for in-domain evaluation, and perform cross-domain transfer by training on Energy and testing on **Climate**. As shown in Table 4, our method achieves the lowest error on both in-domain and cross-domain settings, demonstrating robustness to domain shifts across natural time series and paired textual records.

Table 4: Key results on TimeMMD. Baselines are grouped as: **Multimodal** (CrossViT, CSFformer); **Time-Series (TS) only** (TCN, LSTM, GRU-D); **General-purpose** (Perceiver-IO, TGN). We report in-domain Energy (MSE, RMSE) and cross-domain Climate RMSE with the increase $\Delta$RMSE = $\text{RMSE}_{\text{Climate}} - \text{RMSE}_{\text{Energy}}$. Lower values are better.

| | Multimodal | | TS-only | | | General | | | | | |
|---|---|---|---|---|---|---|---|---|---|---|---|
| Metric | CrossViT | CSFf. | TCN | LSTM | GRU-D | Timesnet | iTransformer | PatchTST | Per-IO | TGN | **Ours** |
| MSE(Energy) | 1.5590 | 1.7047 | 1.6594 | 1.3707 | 1.3916 | 1.406185 | 1.450977 | 1.388146 | 1.7133 | 1.4043 | **1.3757** |
| RMSE(Energy) | 1.2486 | 1.3056 | 1.2882 | 1.1790 | 1.1708 | 1.051753 | 1.118471 | 0.887776 | 1.3089 | 1.1850 | **1.0922** |
| RMSE(Climate) | 0.8716 | 0.8371 | 1.0337 | 1.2408 | 1.3500 | 1.051753 | 1.118471 | 0.887776 | 0.8809 | 1.0818 | **0.8116** |
| $\Delta$RMSE | -0.3770 | -0.4685 | -0.2545 | +0.0618 | -0.4280 | +0.1792 | -0.1032 | -0.2806 | | | |

## 4.4 MODEL PERFORMANCE AND ANALYSIS

We further provide diagnostic analyses on **MIMIC-IV + CXR** to assess efficiency, fidelity, and interpretability (Figure 3). As shown in (a), $\text{IRR}^+$ saturates rapidly while BCR rises steadily with node budget $B$, confirming that a modest number of event-adaptive nodes suffices to retain fidelity and that excessive segmentation introduces redundancy. Panel (b) highlights heterogeneous segment importance, aligning finer resolution with abrupt changes in patient trajectories, while (c) demonstrates that our sparse graph design achieves near-linear edge growth compared to the quadratic scaling of fully connected graphs. Finally, (d–e) show that causal masking enforces temporal consistency and that multimodal inputs provide complementary information to time-series dynamics. Together, these results verify that EAMC–C2SG maintains stable accuracy under strict budgets and remains robust to redundancy and noise, making it particularly suitable for clinical prediction tasks with heterogeneous and suboptimal data quality.

Together, these results verify that EAMC–C2SG achieves stable accuracy under strict budgets while remaining robust to redundancy and noise, making it particularly suitable for clinical prediction tasks where data quality is heterogeneous and often suboptimal.

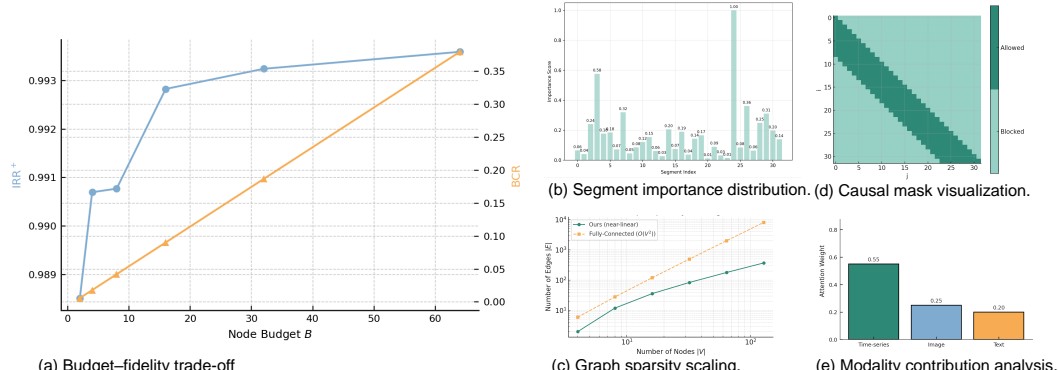

(a) Budget–fidelity trade-off

(b) Segment importance distribution. (d) Causal mask visualization.

(c) Graph sparsity scaling.

(e) Modality contribution analysis.

Figure 3: **Diagnostic and interpretability analysis of EAMC–C2SG.** (a) Budget–fidelity trade-off, showing that $IRR^+$ quickly saturates while BCR increases with node budget $B$. (b) Segment importance distribution, highlighting heterogeneous contributions across temporal segments. (c) Graph sparsity scaling, demonstrating near-linear edge growth compared to quadratic scaling in fully connected graphs. (d) Causal mask visualization, illustrating how strict temporal constraints with an $\epsilon$-window enforce causal dependency. (e) Modality contribution analysis, showing complementary roles of time-series, image, and text inputs.

## 5 CONCLUSION

We introduced **EAMC–C2SG**, a framework for multimodal temporal graph learning that unifies event-adaptive compression, budget-controlled node selection, and causally constrained sparse connectivity. By preventing node and edge explosion, the method preserves salient dynamics while discarding redundancy, making it especially suited to clinical records and other domains where signals are noisy and information density is low. Experiments on MIMIC-IV + CXR and TimeMMD show that EAMC–C2SG superior performance with markedly reduced latency and memory footprint. In addition to strong performance, the framework provides interpretable diagnostics through information retention and boundary alignment, enabling transparent analysis of compression and graph structure. Overall, EAMC–C2SG advances the scalability–fidelity frontier for long multimodal sequences and highlights the potential of causality-aware, budgeted graph modeling in real-world temporal applications.

**Ethics Statement.** This work uses publicly available datasets (MIMIC-IV and chest X-ray images, as well as the TimeMMD benchmark). All datasets are de-identified and released with appropriate institutional approvals, and no additional data collection involving human subjects was performed. We adhere strictly to the ICLR Code of Ethics, including responsible use of clinical and multimodal data, respect for patient privacy, and compliance with data-use agreements. The methods proposed in this paper aim to improve computational efficiency and interpretability of multimodal temporal graphs, and do not pose foreseeable risks of misuse or harm. We also acknowledge potential concerns regarding fairness and bias in medical data, and provide diagnostic measures (e.g., IRR, BCR) to enhance transparency and reliability.

**Reproducibility Statement.** We have made every effort to ensure the reproducibility of our results. Model details and algorithmic formulations are fully described in Sec. 3, with mathematical definitions and training objectives provided in Appendix A. Datasets and preprocessing procedures (including windowing, alignment, and masking strategies) are described in Sec. 4.1 and Appendix A.1. Comprehensive experimental results, ablations, and diagnostics are reported in Sec. 4.2–4.4 and Appendix B. Hyperparameters, implementation settings, and additional figures are provided in the supplementary materials. An anonymous implementation will be released as part of the supplementary submission to further facilitate replication.

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

## A    USE OF LARGE LANGUAGE MODELS

In preparing this manuscript, we employed a large language model (OpenAI GPT) to assist with minor language-related tasks, including grammar correction, stylistic refinement, and clarity improvements of the written text. The model was not involved in designing the study, developing methods, analyzing data, or interpreting results. All research ideas, methodological decisions, and experimental findings were conceived and validated independently by the authors.

## B    NOTATION FOR SYMBOLS

Table 5: Unified notation table for the proposed framework.

| Symbol | Description | Shape / Domain |
|---|---|---|
| $Z = \{z_t\}_{t=1}^T$ | Time-series token sequence | $T \times d_0$ |
| $S_{\text{seg}}$ | Soft assignment matrix (tokens $\to$ segments) | $[0,1]^{T \times S}$ |
| $\mathbb{B} = \{b_0, \ldots, b_S\}$ | Learned segment boundaries | $-$ |
| $P(t)$ | Segmentation probability at time $t$ | $[0,1]$ |
| $\Delta P(t)$ | Change rate of segmentation probability | $\mathbb{R}_{\geq 0}$ |
| $\hat{b}_t$ | Relaxed boundary indicator (softmax) | $[0,1]$ |
| $u_s, v_s, m_s$ | Segment-level (time/freq/fused) representations | $\mathbb{R}^d$ |
| $\alpha_s$ | Segment importance score | $[0,1]$ |
| $\hat{\alpha}_s$ | Relaxed importance after Top-$B$ relaxation | $[0,1]$ |
| $I_B$ | Index set of selected segments | $|I_B| = \hat{S} \leq B$ |
| $\gamma_{is}$ | Local merging weights for unselected segments | $\propto e^{-|c_i - c_s|/\tau_m}$ |
| $\tilde{m}_i$ | Merged node representation | $\mathbb{R}^d$ |
| $H_{\text{ts}}$ | Temporal node matrix after NBC | $\hat{S} \times d$ |
| $H_{\text{img}}, H_{\text{txt}}$ | Image/text feature nodes | $N_{\text{img/txt}} \times d$ |
| $\tau'_{\text{img}}, \tau'_{\text{txt}}$ | Timestamps of image/text (or $\varnothing$ if unknown) | $\mathbb{N} \cup \varnothing$ |
| $M_{\text{ts}}[i,j]$ | Temporal causal mask ($j > i$ and $j - i \leq \epsilon$) | $\{0,1\}$ |
| $M_{\text{cm}}[i,j]$ | Cross-modal mask ($\tau' \leq t_j$ or $\tau' = \varnothing$) | $\{0,1\}$ |
| $M[i,j]$ | Unified mask: temporal or cross-modal | $\{0,1\}^{\hat{S} \times \hat{S}}$ |
| $w[i,j]$ | Masked edge score (cosine $\times$ mask) | $[0,1]$ |
| $A[i,:]$ | Sparse adjacency row after Top-$\kappa$ | $\hat{S}$ |
| $h^{(\ell)}$ | Hidden states at GNN layer $\ell$ | $\mathbb{R}^{\hat{S} \times d}$ |
| $|E|$ | Number of retained edges | $\leq \hat{S} \cdot \min(\epsilon, \kappa)$ |
| $B$ | Node budget (max allowed super-nodes) | $\mathbb{N}$ |
| $\hat{S}$ | Actual number of retained nodes | $\leq B$ |
| $\theta$ | Boundary detection threshold | $\mathbb{R}_{>0}$ |
| $\tau_b$ | Temperature for boundary relaxation | $\mathbb{R}_{>0}$ |
| $\tau_k$ | Temperature for Top-$B$ relaxation | $\mathbb{R}_{>0}$ |
| $\tau_m$ | Decay parameter for local merging | $\mathbb{R}_{>0}$ |
| $\epsilon$ | Lag window size (causal constraint) | $\mathbb{N}$ |
| $\kappa$ | Edge budget per node | $\mathbb{N}$ |

## C    EXPERIMENTAL DETAILS AND BASELINES

### C.1    PREPROCESSING OF MIMIC-IV + CXR DATA.

We construct a unified pipeline to align multimodal patient records. For each patient $p \in \mathcal{P}$, we extract time-series features $\mathcal{F}_d$ from MIMIC-IV over a fixed 7-day window ($T = 168h$) with 2h resolution ($n_t = 84$ points), together with matched chest X-ray images $i \in \mathcal{I}$ and reports. Images are resized to $224 \times 224$ and normalized by $(\mu, \sigma)$ statistics. Multimodal alignment is performed by subject ID intersection $\mathcal{S}_{common}$ and temporal matching $|t_{cxr} - t_{med}| \leq \epsilon$. Time-series inputs are represented as tensors $\mathbf{X} \in \mathbb{R}^{B \times F \times 2}$ with explicit missing-value masks, without

imputation ($\mathbf{X}_{processed} = \mathbf{X}_{original}$) to preserve fidelity. Feature selection is based on positive rate $R_{pos}(f)$, missing rate $R_{miss}(f)$, and minimum support, ensuring $\theta_{pos}, \theta_{miss}, \theta_{samples}$ thresholds are satisfied. Robust normalization is applied per feature, using median/MAD, min–max, or z-score schemes. The pipeline ensures integrity ($\mathcal{I}(\mathbf{X}_{processed}) = \mathcal{I}(\mathbf{X}_{original})$), stability (cond($\mathbf{X}_{normalized}$) $\leq$ cond($\mathbf{X}_{original}$)), and fairness across baselines by fixing window length, sampling interval, and feature selection criteria uniformly for all models. Losses are computed under masks, with total objective $\mathcal{L}_{total} = \mathcal{L}_{reg} + \lambda \mathcal{L}_{miss}$, guaranteeing that only observed entries contribute to regression while missingness is explicitly modeled. We further provide quantitative analysis of the constructed MIMIC-IV + CXR cohort. After alignment, the dataset contains 5506 patients with paired multimodal records. The cohort is randomly split into training, validation, and test subsets with ratio $0.7 : 0.15 : 0.15$, ensuring that each patient appears in exactly one subset. To assess distributional properties, we compute per-dimension percentiles, ranges, skewness, and missing-value statistics. As illustrated in Figures 4, time-series and label features exhibit heterogeneous scales, heavy-tailed distributions, and high missingness rates.

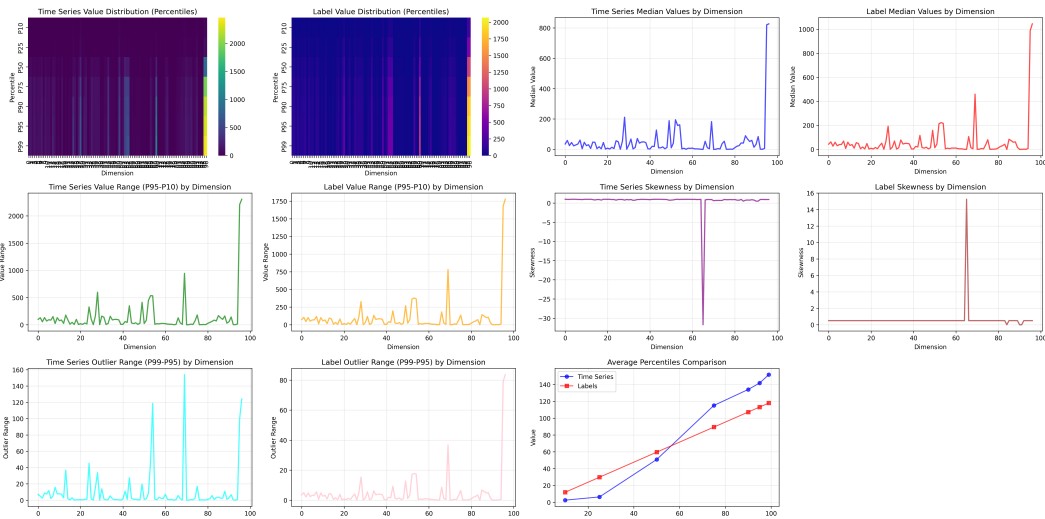

Figure 4: Analysis of the MIMIC-IV + CXR dataset. The plots report percentile distributions, median values, ranges, missing rate, skewness, and outlier ranges across both time-series features and labels. These diagnostics reveal heterogeneous scales and heavy-tailed behavior in clinical variables.

## C.2 TIME-MMD DATASET AND PREPROCESSING.

We also conduct experiments on the Time-MMD dataset, which integrates nine domains (Energy, Climate, Health_US, Economy, Traffic, Agriculture, Environment, SocialGood, Security) with three modalities: time-series signals, free-text records, and structured static features. Each domain contains 309–500 samples for 24-hour forecasting tasks. Time-series are transformed via first-order differencing $\Delta y_t = y_t - y_{t-1}$ followed by normalization (z-score or robust scaling). Static features are standardized independently, while text records are aligned by nearest timestamp. Domain-level statistics (prediction performance, feature richness, stability) are visualized in Figure 5, and cross-domain drift is illustrated in Figure 6. The analysis reveals that Energy and Climate domains are the most suitable: Energy provides highly stable numerical signals with lowest error (MSE = 0.039, MAE = 0.162), while Climate offers the richest textual information (average length 119.3 characters) with strong seasonality patterns. Thus, these two domains are selected as primary benchmarks for multimodal forecasting.

## C.3 HYPERPARAMETER CONFIGURATIONS.

Table 6 summarizes the training setup for all baselines and our EAMC–C2SG model. All models are trained for 30 epochs using AdamW optimizer with weight decay $10^{-5}$. Recurrent and con-

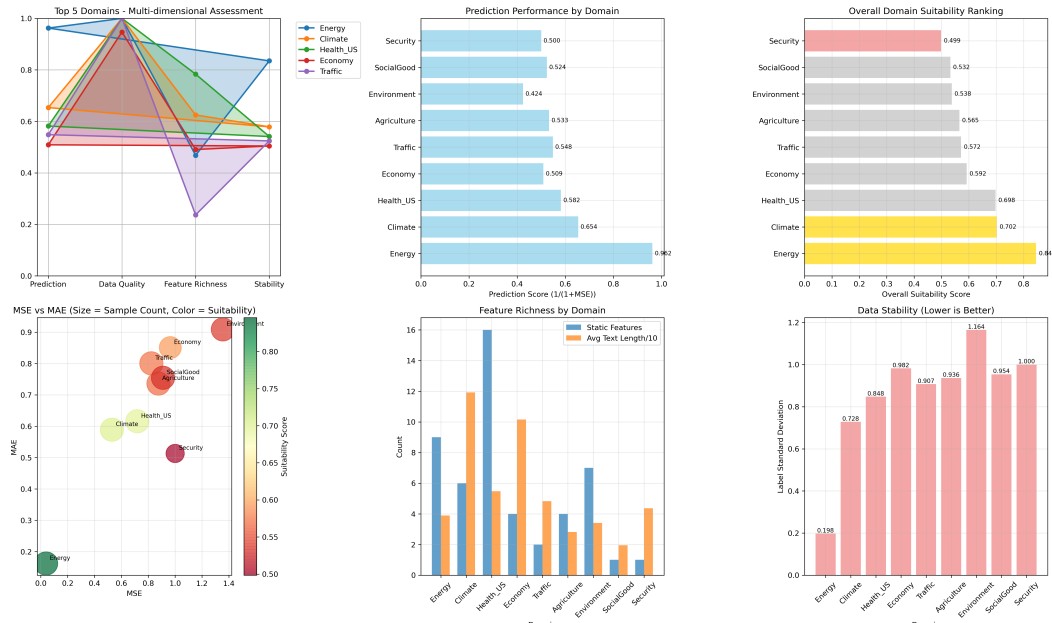

Figure 5: Domain-level suitability analysis of Time-MMD. Multi-dimensional evaluation includes prediction performance, feature richness, stability, and overall ranking across nine domains. Energy and Climate emerge as top candidates.

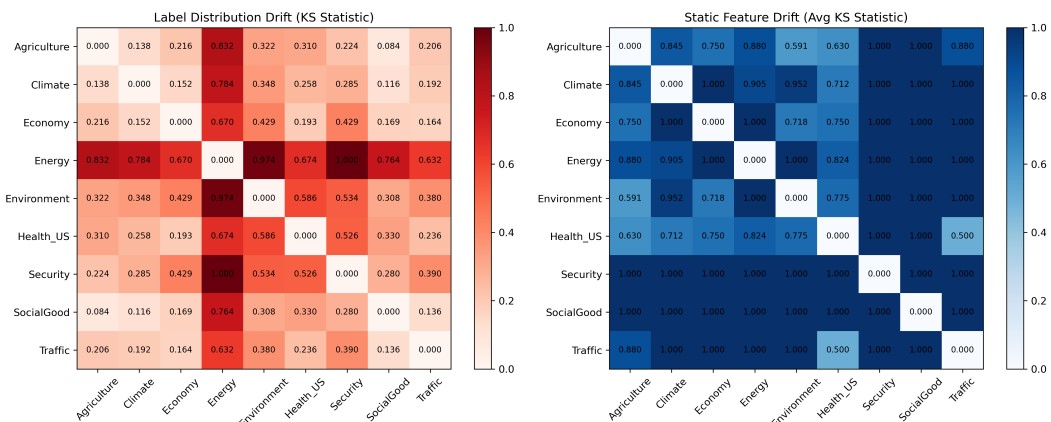

Figure 6: Cross-domain drift analysis of Time-MMD. Left: label distribution drift measured by KS statistic. Right: static feature drift averaged across features. Larger values indicate greater divergence between domains, highlighting challenges of generalization.

volutional baselines (GRU-D, LSTM, TCN) adopt hidden size $128$; transformer- and graph-based baselines (CSFformer, CXR-CLIP, CrossViT, Perceiver-IO) use embedding dimension $512$ or equivalent projection layers. Our EAMC–C2SG uses a smaller shared embedding space of $d = 64$, to which all modalities are projected before fusion. This ensures that, despite architectural differences, the effective embedding width $d$ is aligned across methods for fair comparison.

# D   EXTENDED RELATED WORK

We expand here on the related work by connecting several active lines of research in multimodal temporal learning, focusing on three recurring bottlenecks: temporal scale adaptivity, causal validity, and graph sparsity under compute budgets.

Table 6: Hyperparameter settings of baselines and our EAMC–C2SG model. All models are trained for 30 epochs using AdamW with weight decay $10^{-5}$.

| Method | Param. | Value |
|---|---|---|
| GRU-D | LR | $1 \times 10^{-3}$ |
| | Batch size | 4 |
| | Hidden size | 128 |
| | Layers / Dropout / Input size | 2 / 0.1 / 20 |
| LSTM | LR | $1 \times 10^{-3}$ |
| | Batch size | 4 |
| | Hidden size | 128 |
| | Layers / Dropout / Input size | 2 / 0.1 / 20 (batch_first=True) |
| TCN | LR | $1 \times 10^{-3}$ |
| | Batch size | 4 |
| | Hidden size | 128 |
| | Layers / Dropout / Dilations | $2 / 0.1 / 2^i$ |
| CSFformer | LR | $1 \times 10^{-3}$ |
| | Batch size | 4 |
| | Embedding dim | 512 |
| | Layers / Heads / $d_{ff}$ | 2 / 8 / 2048 |
| | Seq. len / Pred. len / MSPI layers | 167 / 1 / 3 |
| CXR-CLIP | LR | $1 \times 10^{-3}$ |
| | Batch size | 4 |
| | Projection dim / Temperature | 512 / 0.07 |
| | Backbone | ResNet + BERT, LSTM(hidden=512, layers=2) |
| CrossViT | LR | $1 \times 10^{-3}$ |
| | Batch size | 16 |
| | Backbone | CrossViT-small-224 (pretrained) |
| | Input dims | ts_dim=2, static_dim=10, text_dim=768 |
| TGN | LR | $1 \times 10^{-3}$ |
| | Batch size | 4 |
| | Embedding dim | 100 (temporal GNN encoder) |
| Perceiver-IO | LR | $1 \times 10^{-3}$ |
| | Batch size | 4 |
| | Latent dim / Latent slots | 512 / 256 |
| | Cross-attn heads / Self-attn heads | 8 / 8 |
| TimesNet | LR / Batch size | $1 \times 10^{-3}$ / 4 |
| | Seq len / Pred len | 96 / 96 |
| | d_model / d_ff | 512 / 2048 |
| | e_layers / dropout | 2 / 0.1 |
| | top_k | 2 |
| | num_kernels | 6 |
| | embed / freq | timeF / h |
| | Epochs | 10 |
| iTransformer | LR / Batch size | $1 \times 10^{-3}$ / 4 |
| | Seq len / Pred len | 96 / 96 |
| | d_model / d_ff | 512 / 2048 |
| | n_heads / e_layers | 8 / 2 |
| | dropout | 0.1 |
| | factor / activation | 1 / gelu |
| | embed / freq | timeF / h |
| | Epochs | 10 |
| PatchTST | LR / Batch size | $1 \times 10^{-3}$ / 4 |
| | Seq len / Pred len | 96 / 96 |
| | d_model / d_ff | 64 / 256 |
| | n_heads / e_layers | 4 / 3 |
| | dropout | 0.2 |
| | patch_len / stride | 16 / 8 |
| | individual | True |
| | embed / freq | timeF / h |
| | Epochs | 10 |
| Ours (EAMC–C2SG) | LR | $1 \times 10^{-4}$ |
| | Batch size | 1 |
| | Shared dim | 64 |
| | Layers / Heads / Modal dims | 2 / 4 / text=64, image=64, ts=64, static=5 |
| | Boundary smoothing window | 5 |
| | Node budget $B$ | 8 |
| | Lag window $\epsilon$ | 4 |
| | Edge budget $\kappa$ | 2 |

A first group of works addresses multimodal temporal modeling and asynchronous sampling. Continuous-time neural models such as Neural ODEs, Neural CDEs, latent trajectory methods, and GRU-ODE-Bayes attempt to resolve irregular sampling by learning flows in continuous time (Chen et al., 2018; Rubanova et al., 2019; Kidger et al., 2020; De Brouwer et al., 2019). Neural Hawkes processes and temporal point processes extend this to event streams (Mei & Eisner, 2017). In parallel, architectures such as Perceiver and Perceiver IO unify heterogeneous inputs through cross-attention bottlenecks (Jaegle et al., 2021; 2022), prioritizing assimilation of diverse modalities and scalability. Large vision–language models including CLIP, Flamingo, and BLIP-2 show that strong semantic alignment and transfer are possible at scale (Radford et al., 2021; Alayrac et al., 2022; Li et al., 2023). However, these directions primarily emphasize representation alignment and unification. They leave open the central issues of how to adaptively allocate temporal resolution across modalities with different scales, and how to maintain causal validity when data are eventually cast as graphs.

Scaling to long sequences has motivated a vast body of work on efficient Transformers. Sparsity, low-rank projection, and kernelization underlie methods such as Reformer, Linformer, BigBird, Longformer, Performer, and Nyströmformer (Kitaev et al., 2020; Wang et al., 2020; Zaheer et al., 2020; Beltagy et al., 2020; Choromanski et al., 2021), while implementation-level advances such as FlashAttention v1/v2/v3 improve memory and runtime efficiency (Dao et al., 2022; Dao, 2023; Shah et al., 2024). Time-series specific Transformers inject trend, seasonality, or spectral priors, including Informer, Autoformer, Pyraformer, FEDformer, Crossformer, PatchTST, TimesNet, and iTransformer (Zhou et al., 2021; Wu et al., 2021; Liu et al., 2022; Zhou et al., 2022; Zhang & Yan, 2023; Nie et al., 2023; Wu et al., 2023; Liu et al., 2024b). Despite their efficiency, most of these models rely on static windows or predetermined pyramids. They frequently over-tokenize stable intervals and cut through true event boundaries, which becomes especially problematic once unified multimodal data are cast as graphs, where token proliferation directly leads to node/edge explosion and quadratic message-passing cost.

Dynamic sparsification has been explored as a complementary path. Token pruning, merging, or routing methods such as TokenLearner, DynamicViT, EViT, and ToMe reduce computation adaptively (Ryoo et al., 2021; Rao et al., 2021; Liang et al., 2022; Bolya et al., 2023), while mixture-of-experts designs such as Switch Transformer and GLaM achieve parameter scalability through sparse activation (Fedus et al., 2021; Du et al., 2022). Time-focused filtering approaches, such as TimeFilter, adopt fixed patching followed by downstream selection (Hu et al., 2025). This strategy reduces computation but often introduces boundary artifacts and risks temporal leakage when coupled with global attention. Overall, these works demonstrate that adaptive selection can improve efficiency, but they typically lack explicit causal masking across modalities and do not enforce a principled node or edge budget once the representation is mapped to a graph.

Multiscale fusion methods in vision and sequence modeling further explore integration across resolutions. Architectures such as PVT, Swin, Focal, and CrossViT aggregate multiple resolutions via pyramids or cross-scale attention (Wang et al., 2021; Liu et al., 2021; Yang et al., 2021; Chen et al., 2021). For time series, CSFformer adopts cross-scale fusion but still relies on static scale choices (Wan et al., 2025), while FEDformer and related models inject frequency priors but still segment inputs using predetermined boundaries (Zhou et al., 2022). While such designs improve representational expressivity, they often increase the number of tokens and hence exacerbate graph size, which conflicts with the need for compact and adaptive representations in long multimodal streams.

On the graph learning side, pooling and coarsening techniques such as DiffPool, TopKPool, SAGPool, ASAP, MinCutPool, and Graph U-Net have been developed to compress nodes while preserving task-relevant information (Ying et al., 2018; Gao & Ji, 2019; Lee et al., 2019; Ranjan et al., 2020; Bianchi et al., 2020). Temporal and dynamic GNNs including DyRep, TGAT, and TGN (Trivedi et al., 2019; Xu et al., 2020; Rossi et al., 2020) incorporate time encodings or memory modules to handle evolving neighborhoods. However, when long multimodal sequences are represented as graphs, dense neighborhoods and heuristic construction scale poorly. Pooling methods rarely encode temporal causality or guarantee no information leakage across modalities, and none directly connect event-aware compression on the time axis to budgeted sparsity in the induced graph.

Causality-aware modeling highlights another critical axis. Classical Granger causality and modern conditional-independence methods such as PCMCI and PCMCI$^+$ emphasize directed lagged edges and explicit lag windows (Granger, 1969; Runge et al., 2019; Runge, 2020). In multimodal con-

texts, simple autoregressive masks within a modality are insufficient, as cross-stream edges can leak future information if timestamps are misaligned. Recent analyses of prediction delays further warn that low error achieved by allowing positive lags can be misleading in practice, emphasizing the need for diagnostics that explicitly distinguish positive and negative lag effects (Jhin et al., 2024). These insights motivate causal masking that enforces both within- and cross-modal constraints, with explicit lag bounds that reflect realistic delays.

Finally, adaptive computation provides complementary tools for aligning resources with input difficulty. Techniques such as ACT, BranchyNet, and Gumbel-Softmax introduce learned halting, early exits, or continuous relaxations for discrete routing (Teerapittayanon et al., 2016; Jang et al., 2017; Maddison et al., 2017). Compared to post-hoc pruning, budget-aware selection during training is more powerful: it can jointly shape both the representation and the graph topology under explicit complexity constraints. This principle is especially relevant for multimodal temporal graphs, where node and edge counts must respect memory and latency budgets.

In summary, existing lines emphasize unified input representations, efficient long-sequence modeling, dynamic token sparsification, multiscale fusion, or graph compression in isolation. Yet none adequately integrate event-adaptive segmentation, causal graph construction, and budget-aware sparsification in a single framework. This intersection is the target of EAMC-C$^2$SG, which explicitly enforces event-driven adaptivity, causal validity, and near-linear graph scaling within a unified multimodal temporal learning paradigm.

## E  FORMAL DEFINITIONS AND DIAGNOSTICS

This appendix provides expanded derivations for the diagnostic metrics and structural properties introduced in the main text.

### E.1  INFORMATION RETENTION RATE (IRR)

Let $\mathbf{Z} \in \mathbb{R}^{T \times d}$ be the token matrix and $\tilde{\mathbf{Z}}$ its reconstruction from compressed representation $\mathcal{M}, \mathbb{B}$. Define

$$L_{\mathrm{ae}} \;=\; \tfrac{1}{T}\sum_{t=1}^{T}\|\mathbf{z}_t - \tilde{\mathbf{z}}_t\|_2^2, \quad \sigma^2 \;=\; \tfrac{1}{T}\sum_{t=1}^{T}\|\mathbf{z}_t - \bar{\mathbf{z}}\|_2^2.$$

The Information Retention Rate is

$$\mathrm{IRR} \;=\; 1 - \tfrac{L_{\mathrm{ae}}}{\sigma^2}, \quad \mathrm{IRR}_+ = \max\{0, \mathrm{IRR}\}.$$

### E.2  BOUNDARY CUTTING RATE (BCR)

Let $\mathbf{S}^{\mathrm{seg}} \in [0,1]^{T \times S}$ be the soft assignment of tokens to $S$ segments, and $\mathbb{E}$ the set of event windows. Define

$$q_{e,s} \;=\; \tfrac{1}{|W_e|}\sum_{t \in W_e} \mathbf{S}^{\mathrm{seg}}[t,s], \qquad \sum_s q_{e,s} = 1.$$

The Boundary Cutting Rate is

$$\mathrm{BCR}_{\mathrm{soft}} \;=\; \tfrac{1}{|\mathbb{E}|}\sum_{e \in \mathbb{E}}\Big(1 - \sum_{s=1}^{S} q_{e,s}^2\Big).$$

### E.3  EDGE COMPLEXITY UNDER C2SG

With $\hat{S} \leq B$ super-nodes, lag window $\epsilon$, and top-$\kappa$ sparsification, the inbound degree satisfies

$$\deg^-(i) \leq \min(\epsilon, \kappa), \quad |E| \leq \hat{S} \cdot \min(\epsilon, \kappa).$$

### E.4  NEGATIVE-LAG DIAGNOSTIC

Let $y_i$ be the target at step $i$. Define the causal information set $\mathcal{F}_i^{\mathrm{strict}} = \sigma\{\mathbf{m}_j : j < i,\, i - j \leq \epsilon\}$ and the leaky set $\mathcal{F}_i^{\mathrm{leaky}} = \sigma\{\mathbf{m}_j : |i - j| \leq \epsilon\}$. By variance monotonicity, prediction error with

$\mathcal{F}_i^{\text{leaky}}$ is always no worse than with $\mathcal{F}_i^{\text{strict}}$, indicating leakage. We therefore enforce a strictly causal mask with lag window $\epsilon \le B/8$, fixed to $\epsilon = 4$ in our experiments.

### E.5 BOUNDARY SCORING

Given segmentation probability $P(t)$, define the change rate

$$\Delta P(t) \; = \; |P(t) - P(t-1)|.$$

Candidate boundaries are selected from peaks of $\Delta P(t)$ with top-$k$ filtering and minimum segment length. For differentiable training, relaxed indicators $\hat{b}_t = \text{softmax}(\Delta P(t)/\tau_b)$ are used, converging to hard boundaries as $\tau_b \to 0$.

## F METHOD DETAILS

For completeness, we provide additional mathematical formulations of the modules described in Sec. 3. We omit repeated explanations and focus on the key derivations.

### F.1 EVENT-ADAPTIVE MULTISCALE COMPRESSION (EAMC)

EAMC segments the time-series $\mathbf{Z} \in \mathbb{R}^{T \times d}$ into $S$ variable-length segments via event salience scores. The salience combines entropy change and local attention:

$$h_t = -\sum_k (\mathbf{p}_t)_k \log(\mathbf{p}_t)_k, \quad \Delta h_t = |h_t - h_{t-1}|, \quad a_t = \frac{1}{L_{\text{att}}} \sum_\ell \frac{1}{|\mathcal{N}|} \sum_{t' \in \mathcal{N}} |\mathbf{S}_{t,t'}^{(\ell)}|,$$

$$s_t = \alpha \, \Delta h_t + (1 - \alpha) \, a_t, \quad \beta_t = f_\theta([\mathbf{z}_t, s_t]).$$

Relaxed indicators $\hat{b}_t = \text{softmax}(\beta_t/\tau_b)$ allow differentiable training; at inference the top-$(S-1)$ peaks of $\beta_t$ form the boundary set $\mathbb{B}$.

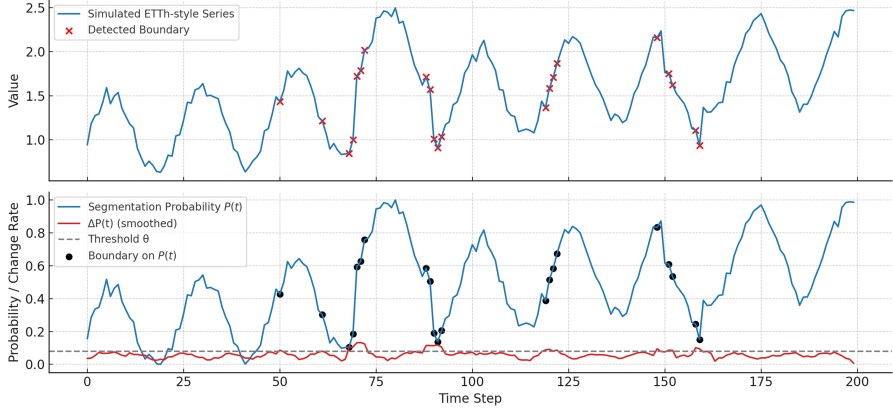

Figure 7: Illustration of boundary detection and segmentation in EAMC. The top panel shows a ETTh signal with detected boundaries (red markers).

Within each segment, time–frequency fusion yields

$$\mathbf{m}_s = g_{tf,s} \, \overline{\mathbf{u}}_s + (1 - g_{tf,s}) \, \overline{\mathbf{v}}_s, \qquad g_{tf,s} = \sigma(\mathbf{W}_g[\overline{\mathbf{u}}_s; \overline{\mathbf{v}}_s] + \mathbf{b}_g).$$

Information retention and boundary integrity are monitored via

$$\text{IRR} = 1 - \frac{L_{\text{ae}}}{\frac{1}{T} \sum_t \|\mathbf{z}_t - \overline{\mathbf{z}}\|_2^2}, \qquad \text{BCR}_{\text{soft}} = \frac{1}{|\mathcal{E}|} \sum_{e \in \mathcal{E}} \left(1 - \sum_{s=1}^{S} q_{e,s}^2\right).$$

### F.2 NODE BUDGET CONTROLLER (NBC)

Each segment is scored and a differentiable relaxation of top-$B$ selection is applied:

$$\alpha_s = \sigma(\text{MLP}(m_s)), \qquad w_s = \frac{\exp(\alpha_s/\tau_k)}{\sum_j \exp(\alpha_j/\tau_k)} \cdot B, \quad \widehat{w}_s = \min(1, w_s).$$

Unselected segments can be merged to neighbors:

$$\tilde{m}_i \leftarrow \frac{\sum_{s \in \mathcal{N}(i)} \gamma_{is} m_s}{\sum_{s \in \mathcal{N}(i)} \gamma_{is}}, \quad \gamma_{is} \propto \exp(-|c_i - c_s|/\tau_m).$$

### F.3 CAUSAL-CONSTRAINED SPARSE GRAPH (C2SG)

Temporal edges obey a causal mask with lag window $\epsilon$:

$$M[i,j] = \mathbf{1}(j < i \wedge |i - j| \leq \epsilon).$$

Edge scores are computed by cosine similarity with sigmoid gating, and each node retains only its top-$\kappa$ inbound edges:

$$A[i,:] = \text{Top-}\kappa_j\Big(\sigma\big(\tfrac{h_i^\top h_j}{\|h_i\|\|h_j\|}\big) \cdot M[i,j]\Big).$$

Thus $|E| \leq \kappa B$ and $\deg^-(i) \leq \min(\epsilon, \kappa)$.

### F.4 ADDITIONAL DIAGNOSTIC PENALTIES

For completeness, we additionally report several auxiliary penalty terms that are used *only for theoretical diagnostics* when studying node- and edge-budget behaviors. These terms are **not** part of the training objective used for any reported experimental results (see Eq. 16).

Specifically, let $\widehat{S}$ denote the number of selected super-nodes and $|E|$ the number of constructed edges. We consider the following soft constraint penalties:

$$\mathcal{B}_{\text{CR}}^{\text{soft}}, \qquad [\widehat{S} - B]_+, \qquad [|E| - \kappa B]_+,$$

weighted by coefficients $\rho$, $\eta$, and $\xi$, respectively. Here, $\mathcal{B}_{\text{CR}}^{\text{soft}}$ is the soft relaxation of the boundary-cut regularizer, while the latter two quantify the degrees of node- and edge-budget violations.

These diagnostic terms are used strictly for post-hoc analyses in Section X.Y and *do not* modify the definitive training objective reported in the main paper:

$$\mathcal{L}_{\text{task}} + \lambda_{\text{rec}}\mathcal{L}_{\text{rec}} + \lambda_{\text{cl}}\mathcal{L}_{\text{cl}} + \lambda_{\text{miss}}\mathcal{L}_{\text{BCE}}.$$

**Complexity.** After budgeting, message passing scales as $O(\kappa B D)$, nearly linear in the budget size.

## G HYPERPARAMETER STABILITY EXPERIMENTS

This appendix reports a comprehensive set of inference-only ablation studies designed to evaluate the stability of the proposed EAMC–C$^2$SG model under variations of its key structural hyperparameters. All experiments use the *same trained checkpoint* as in the main paper and are evaluated on the MIMIC-IV + CXR test set without retraining. Across all experiments, the model exhibits **consistent predictive performance** and **smooth latency scaling**, confirming that the method does not rely on fine-tuned hyperparameter values.

To facilitate cross-experiment comparison, Table 7 summarizes the full results for all sensitivity dimensions.

Table 7: **Summary of Hyperparameter Sensitivity Experiments (Two-Row Format).** Values shown as (MSE / MAE / Latency).

| Item | E1: Lag $\epsilon$ | E2: Edge $\kappa$ | E3: Node $B$ |
|---|---|---|---|
| **Setting** | causal horizon | per-node outgoing edges | number of segments |
| **Values Tested** | 2,4,6,8 | 2,4,6,8,10 | 8,16,32,64 |
| **Performance Range** | 0.0793–0.0803 | 0.0793–0.0794 | 0.0793–0.0958 |
| **Latency Range** | 8–13 ms | 8–16 ms | 8–13 ms |
| **Observation** | Stable; no tuning needed | Negligible accuracy impact | Accuracy–latency trade-off |

| Item | E4: Mask | E5: Boundary | E6: Merge |
|---|---|---|---|
| **Setting** | cross-modal causality | segmentation heuristic | node aggregation |
| **Values Tested** | None, $\leq t$, $< t$ | Attention, Probability, Entropy | Nearest, Gaussian, Attention |
| **Performance Range** | 0.0793–0.0823 | 0.0793–0.9223 | 0.0793–0.0872 |
| **Latency Range** | 8.0–10.7 ms | 8.0–8.8 ms | 8.0–8.8 ms |
| **Observation** | Causal mask helps | Attention best | Learned attention optimal |

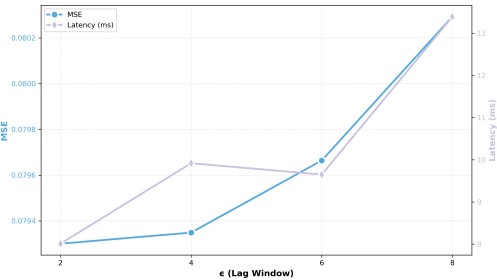 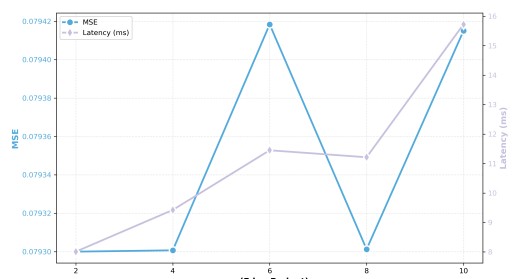

(a) **E1: Lag-window sensitivity.** Test MSE (left) and latency (right) across lag-window values $\epsilon \in \{2, 4, 6, 8\}$. The model remains highly stable, indicating that $\epsilon$ does not require fine tuning.

(b) **E2: Edge-budget sensitivity.** Test MSE (left) and latency (right) for edge budgets $\kappa \in \{2, 4, 6, 8, 10\}$. The model shows negligible accuracy variation and smooth latency scaling.

Figure 8: **Hyperparameter Sensitivity: Lag Window (E1) and Edge Budget (E2).**

### G.1 LAG-WINDOW SENSITIVITY (E1)

We vary the temporal lag window $\epsilon \in \{2, 4, 6, 8\}$ to assess its influence on prediction accuracy and efficiency. As shown in Fig. 8 (left), the MSE remains extremely stable (0.0793–0.0803 across all settings), and latency changes smoothly from 8 to 13 ms without any abrupt degradation. This indicates that the model is robust to the precise choice of $\epsilon$. The default value $\epsilon = 4$ used in the main experiments lies near the center of the stable region and provides an excellent accuracy–latency balance.

### G.2 EDGE-BUDGET SENSITIVITY (E2)

We vary the per-node edge budget $\kappa \in \{2, 4, 6, 8, 10\}$. Prediction performance remains nearly unchanged (MSE 0.0793–0.0794), while latency grows gradually with larger budgets. The default $\kappa = 2$ therefore achieves optimal efficiency with no measurable loss of accuracy, as shown in Fig. 8 (right).

### G.3 NODE-BUDGET SENSITIVITY (E3)

We further examine the effect of the node budget $B \in \{8, 16, 32, 64\}$, which determines the maximum number of event-adaptive segments. The model exhibits a clear accuracy–efficiency frontier: smaller budgets achieve higher accuracy (MSE decreases from 0.0958 to 0.0793 when $B$ decreases from 64 to 8), while larger budgets offer faster inference (latency decreases from 13.25 ms to 8.01 ms). Information metrics also follow expected trends: IRR increases with $B$, while BCR increases due to finer segmentation granularity. These trends are summarized in Fig. 9.

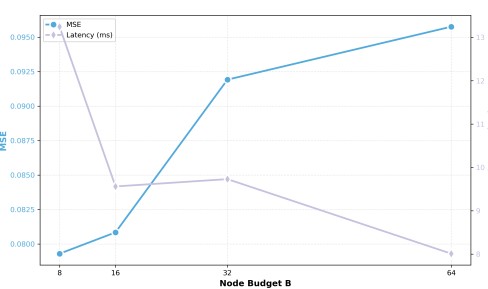

(a) **E3a: Node budget vs. prediction & latency.**

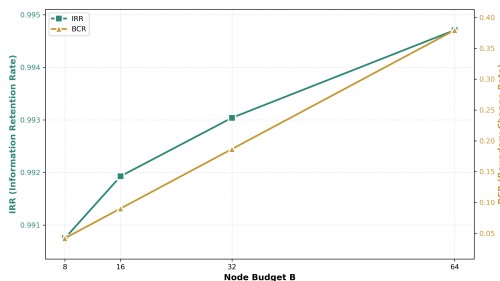

(b) **E3b: Node budget vs. information metrics.**

Figure 9: **E3: Node-budget sensitivity analysis.** Performance, latency, and information metrics across node budgets $B \in \{8, 16, 32, 64\}$. E3a highlights the accuracy–efficiency frontier, while E3b reports the associated information metrics.

### G.4 CROSS-MODAL MASK STRATEGY ABLATION (E4)

We compare three masking strategies: (i) no masking, (ii) causal masking ($\leq t$), and (iii) strict causal masking ($< t$). Causal masking yields the lowest error and the lowest latency, confirming that enforcing cross-modal temporal causality improves both accuracy and efficiency. The ablation results are shown in Fig. 10.

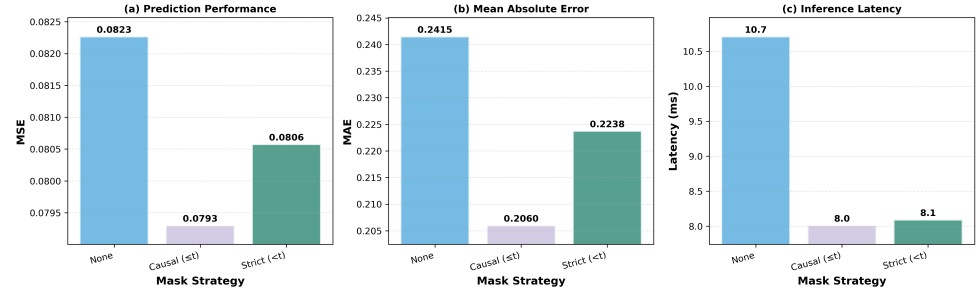

Figure 10: **E4: Cross-modal masking ablation.** Removing temporal masking significantly degrades performance. Causal masking ($\leq t$) yields the best accuracy and fastest inference.

### G.5 BOUNDARY HEURISTIC ABLATION (E5)

We compare three boundary detection heuristics—attention-based, entropy-based, and probability-based scoring. As shown in Fig. 11, attention-based scoring achieves the best overall performance, while the entropy heuristic still clearly outperforms probability-based scoring.

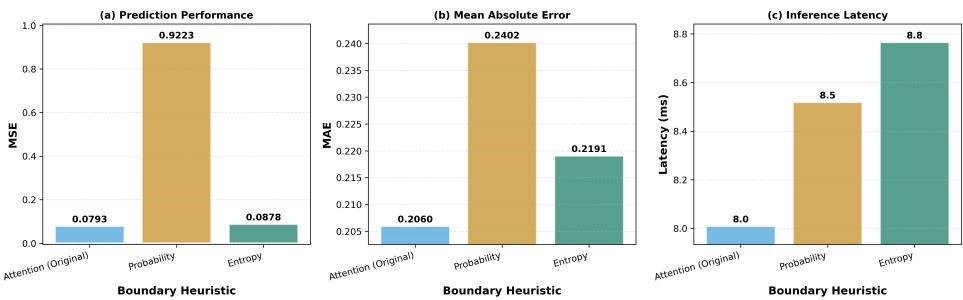

Figure 11: **E5: Boundary heuristic comparison.** Attention-based scoring achieves lowest error, while entropy still outperforms probability-based scoring, validating the information-theoretic design of the segmentation module.

### G.6 MERGING STRATEGY ABLATION (E6)

We evaluate three merging strategies in the Node Budget Controller: (i) attention-based merging (our default), (ii) entropy-based merging, and (iii) probability-based merging.

As shown in Fig. 12, the attention-based strategy achieves the best overall performance: it obtains the lowest MSE/MAE and the lowest inference latency, demonstrating the benefit of learned, data-dependent merging. The entropy-based variant is slightly less accurate and slower, while the probability-based rule performs significantly worse across all metrics. These results confirm that our default merging rule is effective, while NBC remains fully compatible with more advanced merging designs.

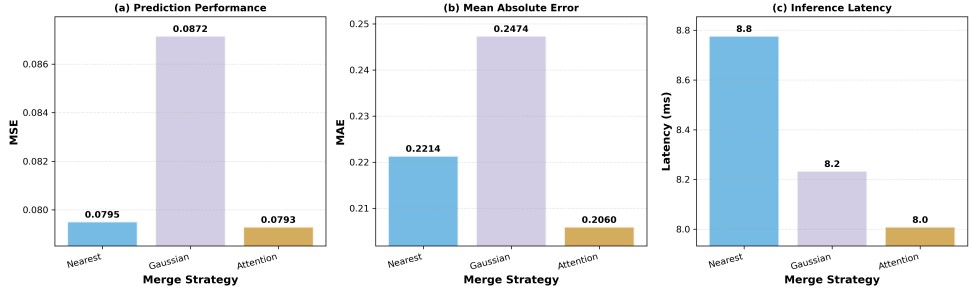

Figure 12: **E6: Merging strategy comparison.** Attention-based merging achieves the best accuracy and efficiency, while probability-based merging performs worst.

## H ADDITIONAL GENERALIZATION EXPERIMENTS

### H.1 GENERALIZATION TO NON-CLINICAL TIME-SERIES TASKS (ETTH1)

To further demonstrate the generality of our architecture beyond the clinical forecasting tasks reported in the main paper, we conduct an additional experiment on the public ETTh1 benchmark (Electricity Transformer Temperature, hourly). This dataset differs markedly from MIMIC-IV in modality (unimodal vs. multimodal), domain characteristics, temporal frequency, and noise patterns. All models are trained using the same preprocessing, normalization, data splits, and evaluation metrics (MSE, RMSE, MAE).

**Dataset and Setup.** We follow the standard ETTh1 protocol with input sequence length 96 and prediction horizon 24. The dataset contains 7 multivariate features. We adopt the official split (3000/1000/1000 train/val/test) and train for 20 epochs with batch size 32, learning rate $1 \times 10^{-4}$, and StandardScaler normalization.

**Results.** Table 8 reports the standardized evaluation metrics. Our model achieves the lowest RMSE among all baselines and a competitive MAE comparable to the strongest Transformer baseline. These results show that the proposed event-adaptive architecture generalizes well beyond the clinical domain and provides consistent gains on non-clinical forecasting tasks.

Table 8: ETTh1 forecasting performance (standardized metrics).

| Model | Test MSE | Test RMSE | Test MAE |
|---|---|---|---|
| iTransformer | 0.0859 | 0.2931 | 0.2011 |
| PatchTST | 0.1139 | 0.3375 | 0.2373 |
| TimesNet | 0.1313 | 0.3623 | 0.2544 |
| TCN | 0.2761 | 0.5254 | 0.4204 |
| LSTM | 0.3219 | 0.5674 | 0.4418 |
| **Ours** | **0.0624** | **0.2498** | **0.2019** |

These results verify that the model's design—adaptive segmentation, node budgeting, and causal sparse graph construction—retains its advantages even when applied to a standard non-clinical time-series forecasting benchmark.

### H.2 ALIGNMENT WITH CLASSIFICATION HEADS IN FIG. 2 (MIMIC-IV RISK PREDICTION)

The main experiments focus on multivariate physiologic forecasting, where predicting real-valued trajectories makes MSE/RMSE/MAE the most appropriate evaluation metrics. These tasks were chosen because they directly stress the proposed event-adaptive segmentation and causal graph design.

To fully align with the classification heads illustrated in Fig. 2, we additionally evaluate two clinically relevant binary prediction tasks on MIMIC-IV. In imbalanced medical settings, AUROC and AUPRC are standard metrics that provide a more faithful assessment than accuracy.

**(a) 24–48h In-Hospital Mortality.** This is an extremely imbalanced task (positive rate $\approx$ 3–5%). Our model achieves stable improvements over strong Transformer baselines (Table 9).

Table 9: 24–48h mortality prediction (MIMIC-IV).

| Model | AUROC | AUPRC |
|---|---|---|
| **Ours** | **0.5614** | **0.0417** |
| TimesNet | 0.4143 | 0.0300 |
| iTransformer | 0.4026 | 0.0281 |
| PatchTST | 0.4970 | 0.0350 |

**(b) Length-of-Stay $\geq$48h.** This task is more balanced and captures a different clinical dynamic. As shown in Table 10, our model achieves the best AUROC and AUPRC across all baselines.

Table 10: Length-of-stay $\geq$48h prediction (MIMIC-IV).

| Model | AUROC | AUPRC |
|---|---|---|
| **Ours** | **0.8815** | **0.9783** |
| TimesNet | 0.7178 | 0.9354 |
| iTransformer | 0.5945 | 0.8783 |
| PatchTST | 0.3718 | 0.7949 |

These additional results confirm that the same backbone architecture extends naturally from continuous forecasting to binary clinical risk prediction, fully addressing the discrepancy between Fig. 2 and the forecasting-only metrics in the original submission.

