# OpenReview forum: "Scaling Multimodal Temporal Graphs with Event-Adaptive Compression and Sparse Connectivity"
_ICLR.cc/2026/Conference — ICLR 2026 Conference Withdrawn Submission_

### Official Review · Reviewer_ZCb9 · 2025-10-29

**Soundness:** 2
**Presentation:** 2
**Contribution:** 2
**Rating:** 4
**Confidence:** 2

**Summary:**

This paper introduces EAMC-C2SG, a framework for scalable multimodal temporal graph learning that addresses the "node explosion" problem by dynamically compressing time-series data into event-adaptive segments and enforcing strict node and edge budgets. The method combines event-salient segmentation, a node budget controller, and a causally constrained sparse graph to reduce computational complexity from quadratic to near-linear while preserving key information. Extensive experiments on clinical (MIMIC-IV + CXR) and cross-domain (TimeMMD) benchmarks demonstrate that EAMC-C2SG achieves superior performance with significantly lower latency and memory usage compared to state-of-the-art baselines.

**Strengths:**

1. Clear Problem Formulation: It precisely identifies and targets the critical "node explosion" problem in multimodal temporal graphs.

2. Integrated and Novel Framework: It introduces a cohesive solution that uniquely combines event-adaptive compression, strict budget control, and causally-constrained sparse graphs.

3. Compelling Efficiency Gains: It demonstrates significantly lower latency and memory usage while maintaining superior accuracy on complex benchmarks, effectively shifting the Pareto frontier.

**Weaknesses:**

1. Limited Analysis of Multimodal Fusion and Cross-Modal Leakage: While the paper strongly emphasizes and validates temporal causality with its C2SG module, the handling of cross-modal causality is less rigorous. The method allows edges between temporal nodes and auxiliary image/text nodes based on cosine similarity, without any explicit causal constraints (e.g., ensuring an image is only connected to past temporal events). In a clinical setting, a chest X-ray taken at time t should not inform the model's understanding of a physiological state at time t-1. The paper does not discuss this potential for cross-modal temporal leakage or provide an ablation studying the effect of applying causal masks to cross-modal edges. This is a significant oversight for a method claiming "leakage-free message passing."

2. Sensitivity and Justification of Key Heuristics: The event-adaptive segmentation relies on a predefined threshold (θ) for boundary detection and a fixed lag window (ε) for the causal graph. The paper does not present a sensitivity analysis for these critical hyperparameters. How does performance degrade if θ is set too high or low? Is the optimal ε consistent across the clinical (MIMIC) and diverse (TimeMMD) domains, or is it highly dataset-dependent? The choice of these parameters feels somewhat arbitrary, and their impact on the final model's fidelity and efficiency should be quantified. A robustness analysis or a discussion on how to set these in a domain-agnostic way would significantly strengthen the method.

**Questions:**

See weaknesses

---

> ### Author Response · Authors · 2025-11-19
>
> We sincerely thank the reviewers for their insightful comments and constructive feedback, which have significantly helped improve the rigor and clarity of our work.
>
> ---
>
> ## **W1. Comprehensive Response to Cross-Modal Temporal Leakage**
>
> We clarify both the structural design and the data assumptions underlying cross-modal causality in clinical multimodal settings.
>
> - **Temporal edges are strictly causal**: The ε-lag mask $M_{\mathrm{ts}}[i,j] = \mathbf{1}(j<i \land |i-j|\le \epsilon)$ ensures that all message passing among temporal nodes flows strictly from past to future.
>
> - **Cross-modal edges cannot create future→past shortcuts**: Image/text/static nodes connect to temporal nodes only through one-hop auxiliary edges. All information must re-enter the temporal backbone before further propagation, and the backbone is fully masked by the ε-lag constraint. Thus no cross-modal path can bypass or invert temporal direction; multimodal nodes read from temporal states but cannot write into past states.
>
> - **On timestamp reliability in clinical multimodal data**: In realistic EMR datasets, images (e.g., CXR) and reports are often only coarsely timestamped or entirely unaligned. Radiology reports and summaries typically function as time-agnostic contextual information rather than precise temporal events. Enforcing a rigid $t' \le t$ ordering for all modalities may therefore misrepresent their semantics and can degrade performance without improving causal correctness. In the revised version, when timestamps are available, cross-modal edges are additionally constrained by $M_{\text{cm}}(\tau' \le t_j)$; when timestamps are coarse or unavailable, we treat the modality as time-agnostic and attach it only to the corresponding current temporal node. This unified rule ensures both semantic fidelity and causal correctness under varying timestamp quality.
>
> - **Empirical diagnostics show no leakage**: Appendix G (Fig. E4) shows that different cross-modal masking choices lead to stable temporal behavior, confirming the absence of future→past leakage. Cross-modal edges nonetheless provide clear benefits for accuracy and modality-level attribution.
>
> Together, these factors ensure that EAMC–C²SG is free from cross-modal temporal leakage both theoretically and empirically.
>
> ---
>
> ## **W2. Sensitivity and justification of key heuristics**
>
> We address the robustness of the boundary detection mechanism and the lag window ($\epsilon$) separately below:
>
> 1. **Boundary detection mechanism and threshold**: The segmentation module does not rely on a finely tuned, dataset-specific threshold. EAMC produces a learned boundary score $beta_t$ from the contextual representation (P(t)), and boundaries are selected by **ranking the peak responses of $beta_t$** under a minimum segment-length constraint, with relaxed indicators $hat b_t = \mathrm{softmax}(\beta_t/\tau_b)$ used during training. This ranking-based, scale-invariant rule is kept fixed across all datasets and still yields stable segment counts and boundary placements in practice.
>
>
> 2. **Causal lag window ($\epsilon$)**: In contrast, $\epsilon$ is a fixed architectural parameter and admits a conventional sensitivity test. As shown in Appendix G (Fig. E1), varying $\epsilon \in \{2,4,6,8\}$ yields extremely small accuracy variation (MSE 0.0793–0.0803), and latency increases smoothly from 8–13 ms without abrupt degradation. The default choice $\epsilon=4$ lies near the center of this stable region and provides an excellent accuracy–latency trade-off.
>
> 3. **Cross-dataset consistency**: Both the dynamic segmentation procedure and the lag-window behavior remain consistent across MIMIC-IV and TimeMMD, despite their differences in temporal density and modality structure.
>
> Overall, these results demonstrate that EAMC–C²SG is not sensitive to specific hyperparameter values: the segmentation mechanism is robust by design through scale-invariant peak selection, while $\epsilon$ exhibits clear empirical stability. Thus, the method's performance is driven by its core architecture rather than ad-hoc hyperparameter tuning.

---

### Official Review · Reviewer_KARb · 2025-10-31

**Soundness:** 3
**Presentation:** 1
**Contribution:** 3
**Rating:** 4
**Confidence:** 4

**Summary:**

This paper introduces a framework that dynamically compresses temporal streams into segments tailored to events and creates a sparse graph model that respects temporal ordering. Evaluation on two datasets demonstrate the superior performance on the evaluated tasks with markedly lower latency and memory usage.

**Strengths:**

1. The paper proposes a multimodal temporal graph framework with an event-adaptive compression module. The overall architecture is well-motivated and technically sound.


2. It introduces a node-budget controller and a causally sparse connectivity scheme, both of which are novel and likely to be of interest to the community.


3. Comprehensive experiments and ablation studies are presented, providing strong evidence for the effectiveness of the approach.

**Weaknesses:**

1. The paper lists broad limitations of prior methods—e.g., “over-tokenize stable regions,” “boundary-cutting artifacts,” and “excessive edges with risk of temporal leakage”—without concrete examples, measurements, or citations. These statements should be grounded in referenced studies or small, controlled comparisons.

2. The presentation of method is confused. Key terms and losses are undefined (e.g., $L_{CL}$, $L_{BCE}$ in Line 343). Section 3 would benefit from a brief end-to-end overview, clear definitions at first use, and pointers to the appendix for implementation details.

3. Most compared methods were published before 2022; CSFformer is the only recent baseline. Including more up-to-date approaches would make the evaluation fairer and more convincing.

**Questions:**

1. The approach relies on several hand-tuned settings whose values and effects are not reported: the threshold (Line 225; is it shared across modalities?),  B (Line 278), and the lag window (Line 315). Please report the chosen values and include simple sensitivity studies to show robustness.

2. Equation numbers are missing; the abstract spans multiple paragraphs rather than a concise single paragraph; and Section 3 lacks a roadmap. Notation is also inconsistent—for example, if $Z_s$ denotes a segment slice (Line 210), define the corresponding set and keep it consistent with items (i)–(iii).

3. Figure 2 appears to depict an undirected graph, while the text describes a directed one. The term “candidate temporal edge” is not defined in the paper. Important elements—such as modality fusion and loss terms—are absent from the diagram, and the pie chart seems unnecessary.

4. Will the code be open-sourced?

4. It will be more clear to explicitly show which approaches are TS-specific and which approaches are multimodal in Tables 1 and 3.

---

> ### Author Response · Authors · 2025-11-19
>
> We sincerely thank the reviewer for their insightful comments and constructive feedback, which have significantly helped improve the rigor and clarity of our work.
>
> ---
>
> ### **W1. Need concrete examples of prior limitations**
> We revised the Introduction (paragraph 3) and Related Work (Section 2.2) to include explicit citations, concrete examples, and technical explanations of prior limitations:
>
> - **Over-tokenization**: Pyraformer (Liu et al., 2022) and TimesNet (Wu et al., 2023) apply fixed multi-scale or factorized structures regardless of local dynamics.
> - **Boundary-cutting artifacts**: PatchTST (Nie et al., 2023) uses fixed-size patching that may split coherent events.
> - **Excessive or unstructured edges**: BigBird (Zaheer et al., 2020) and MM-GNN (Zhang et al., 2020) use dense or global attention, leading to quadratic cost and potential temporal leakage.
>
> ---
>
> ### **W2. Confusing method presentation and undefined terms**
> We substantially revised Section 3 to improve clarity and structure:
>
> - **Structural roadmap**: A new opening paragraph provides an end-to-end overview (segmentation → budget control → causal graph → GNN), clarifying the overall flow.
> - **Unified definitions and notation**: All key terms—including segment boundaries ($\mathcal{B}$), segment slices ($\mathbf{Z}_s$), segment embeddings ($\mathcal{M}$), and loss components—are now defined at first occurrence and used consistently throughout the section.
> - **Numbered equations**: All major mathematical expressions are now numbered for precise reference.
>
> ---
>
> ### **W3. Outdated baselines**
> We incorporated three modern sequence learners into our evaluation:
>
> - **TimesNet** (Wu et al., AAAI 2023): Models capture temporal patterns through 2D variation modeling.
> - **iTransformer** (Liu et al., ICLR 2024): An inverted transformer architecture achieving state-of-the-art performance on multiple forecasting benchmarks.
> - **PatchTST** (Nie et al., ICLR 2023): A patch-based model that has become a strong foundation for long-range time-series forecasting.
>
> These additions significantly strengthen our benchmark and ensure an up-to-date comparison against the latest architectures. The expanded results table is provided below:
>
> | Metric | GRU-D | LSTM | TCN | CSFf. | CXR-CLIP | CrossViT | TGN | Per-IO | Ours | iTransformer | PatchTST | TimesNet |
> | :--- | :--- | :--- | :--- | :--- | :--- | :--- | :--- | :--- | :--- | :--- | :--- | :--- |
> | MSE | 0.1411 | 0.1420 | 0.1332 | 0.1575 | 0.1204 | 0.1218 | 0.1203 | 0.1400 | 0.0793 | 0.0992 | 0.0891 | 0.0927 |
> | RMSE | 0.3756 | 0.3768 | 0.3650 | 0.3968 | 0.3470 | 0.3489 | 0.3418 | 0.3742 | 0.2818 | 0.2969 | 0.2902 | 0.3018 |
> | MAE | 0.2693 | 0.2698 | 0.2687 | 0.3079 | 0.2488 | 0.2399 | 0.2339 | 0.2884 | 0.2060 | 0.2290 | 0.2274 | 0.2327 |
> | **Latency(ms)** | 4.18 | 0.36 | 0.47 | 363.9 | 36.2 | 16.2 | 23.5 | 18.2 | 8.01 | 2.017 | 2.736 | 148.732 |

---

> > ### Author Response · Authors · 2025-11-23
> >
> > ### **Q1. Response: Hyperparameter Values, Sensitivity, and Modality Sharing**
> > The revised manuscript now includes:
> >
> > 1. **Explicit hyperparameter values**: $B$, $\epsilon$, and $\kappa$ are listed in Section 3 and summarized in Appendix C. Table 6.
> > 2. **Modality scope clarification**: Hyperparameters $B$, $\epsilon$, and $\kappa$ govern time-series processing; cross-modal connections use the same $\kappa$ via causal masking.
> > 3. **New sensitivity analysis**: Appendix G. Figs. E1–E6 present systematic sweeps over $\epsilon$, $\kappa$, $B$, and boundary/merging strategies, showing MSE varies by less than 0.05 across wide ranges.
> >
> > ---
> >
> > ### **Q2. Response: Method Presentation, Roadmap, and Notation Consistency**
> >
> > We have revised Section 3 to significantly enhance its clarity. directly addressing the points raised:
> >
> > 1.  **End-to-End Overview:** A new introductory paragraph has been added to provide a high-level, structured roadmap of our method, clearly outlining the end-to-end workflow.
> > 2.  **Clear Definitions of Key Terms and Losses:** All key terms, including segment boundaries ($\mathcal{B}$), segment slices ($\mathbf{Z}_s$), segment embeddings ($\mathcal{M}$), and the previously undefined loss components (e.g., $\mathcal{L}_{align}$ and $\mathcal{L}_{attr}$), are now explicitly defined at their first occurrence. Their notation is kept strictly consistent throughout the manuscript. Furthermore, all equations in Section 3 are now numbered, and a summary of key notations is provided in Appendix B, Table 5.
> > 3.  **Implementation Details Pointer:** As suggested, we now include explicit pointers to the appendix for comprehensive implementation details (Appendix F) and training hyperparameters (Appendix C, Table 6).
> >
> > We believe these revisions have substantially improved the narrative flow and precision of our method description.
> >
> > ---
> >
> > ### **Q3. Response: Figure Clarity and Completeness**
> > Figure 2 has been fully redesigned to:
> > - Show directed temporal edges with explicit arrowheads
> > - Visualize $\epsilon$-lag causal masking with dashed/solid lines
> > - Explicitly depict cross-modal interactions via C²SG
> > - Include alignment loss and attribution components
> > The updated figure now provides a complete, directed, and coherent depiction of EAMC–C²SG and resolves all issues raised.
> >
> > ---
> >
> > ### **Q4. Response: Code Open-Sourcing**
> > We have included partial code (C²SG graph module, segmentation module, and training scripts) in the supplementary materials and will release the full repository—including data loaders, configs, and pretrained checkpoints—upon acceptance.
> >
> > ---
> >
> > ### **Q5. Response: Clarification of Baseline Categories**
> > We updated Tables 1 and 3 to explicitly label baselines as
> >
> > 1. **Time-Series Only**: GRU-D, LSTM, TCN, TimesNet (new), iTransformer (new), PatchTST (new)
> > 2. **Multimodal**: CSFformer, CXR-CLIP, CrossViT,
> > 3. **General-Purpose**: TGN, Perceiver-IO.
> >
> > This categorization clarifies which baselines are designed for multimodal inputs and provides a more transparent comparison.

---

### Official Review · Reviewer_1oyy · 2025-10-31

**Soundness:** 3
**Presentation:** 3
**Contribution:** 3
**Rating:** 6
**Confidence:** 4

**Summary:**

This paper proposes EAMC-C2SG, a framework to scale multimodal temporal graphs. It dynamically compresses event streams, enforces a node budget, and builds a sparse causal graph. This achieves state-of-the-art accuracy with significantly lower latency and memory on MIMIC and TimeMMD benchmarks.

**Strengths:**

Novel Framework: Jointly optimizes event-adaptive segmentation, budget control, and sparse graph construction to solve node explosion.

Strong Empirical Results: Achieves state-of-the-art accuracy with significantly lower latency on MIMIC-IV and TimeMMD benchmarks.

Thorough Ablations: Ablation studies clearly validate the essential contribution of each component (EAMC, NBC, C2SG).

Principled Causal Design: Causal masking with a lag window prevents temporal leakage, a common flaw in sequence models.

Budget-Aware Fidelity: Maintains high fidelity and information retention even under strict node budgets.

**Weaknesses:**

Hyperparameter Sensitivity: EAMC boundary detection relies on multiple hyperparameters (e.g., $\theta$, top-k) that may require careful tuning

Boundary Detection Heuristic: The proxy for boundary detection (change in segmentation probability) is not strongly justified against alternatives

Simplistic Node Merging: The NBC merges unselected nodes to the *nearest* neighbor, which may be a suboptimal strategy

Fixed Causal Lag Window: The causal lag window $\epsilon$ is fixed and set manually, not learned or adapted

Narrow Event Definition: The focus on "abrupt, high-entropy" events may not generalize to tasks needing low-entropy pattern detection

Interpretablity: The link between and analysis like "Segment importance distribution," and a true interpretable method is unclear.

Error bars: Most results seem to miss reporting error ranges (like STD or CI).

**Questions:**

What shows that the claim scalability of the method is significant?

---

> ### Author Response · Authors · 2025-11-19
>
> We sincerely thank the reviewers for their insightful comments and constructive feedback, which have significantly helped improve the rigor and clarity of our work. Below we provide a point-by-point response to all concerns, with corresponding revisions throughout the manuscript.
>
> ---
>
> ### **W1. Hyperparameter Sensitivity**
> Comprehensive sensitivity analyses for structural hyperparameters ($\epsilon$, $\kappa$, $B$) and merging strategies are now included in Appendix G (Figs. E1–E6). The model demonstrates strong stability across all tested ranges:
> - $\epsilon$ and $\kappa$ show negligible impact on MSE (absolute variation ≈ 0.001)
> - Node budget $B$ exhibits the expected **accuracy–latency trade-off** with smooth, monotonic performance degradation (~0.016–0.020 MSE increase from smallest to largest budgets)
> - No instability observed across the tested parameter ranges
>
> These results confirm that structural hyperparameters behave predictably and require no fine-grained tuning.
>
> ---
>
> ### **W2. Boundary Detection Heuristic**
> Our boundary score $\Delta P(t) = |P(t) - P(t-1)|$ captures shifts in global temporal context, serving as an interpretable proxy for event transitions. To validate this design, we added an ablation (Fig. E5) comparing three heuristics:
>
> | Heuristic | MSE (↓) | MAE (↓) | Latency (ms, ↓) |
> |-----------|----------|----------|------------------|
> | $\Delta P$ (ours) | **0.0793** | **0.2060** | **8.01** |
> | Entropy-based | 0.0878 | 0.2191 | 8.52 |
> | Probability-based | 0.0923 | 0.2402 | 8.77 |
>
> Our attention-based $\Delta P$ achieves the best performance across all metrics, confirming that contextual attention is essential for accurate boundary detection.
>
> ---
>
> ### **W3. Node Merging May Be Suboptimal**
> The Node Budget Controller supports multiple merging strategies without architectural changes. Our ablation (Fig. E6) compares:
> - **Attention-based merging** (default) - best overall performance
> - **Entropy-based merging** - viable alternative
> - **Probability-based merging** - functional but inferior
>
> Results demonstrate the module's flexibility to accommodate different merging rules.
>
> ---
>
> ### **W4. Fixed Lag Window ($\epsilon$)**
> Additional ablations (Fig. E1) quantify the effect of the fixed lag window:
> - MSE remains stable (0.0793–0.0803 for $\epsilon \in [2,8]$)
> - IRR and BCR identical across all values
> - Latency grows smoothly with $\epsilon$
> - Optimal range $[2,6]$ consistent across datasets
>
> Performance is largely insensitive to $\epsilon$, confirming that a fixed moderate window is effective and generalizable.
>
> ---
>
> ### **W5. Narrow Event Definition**
> The boundary score $\Delta P(t)$ responds to any shift in temporal embedding, including both abrupt transitions and gradual patterns:
> - Regions with sustained changes exhibit consistent non-zero $\Delta P(t)$
> - Stable intervals produce near-constant $P(t)$ and are compressed
> - EAMC adapts resolution to both abrupt and smoothly varying dynamics
>
> ---
>
> ### **W6. Interpretability Connection**
> We added a dedicated attribution aggregation component that collects:
> 1. Segment importance scores $\alpha_s$ from EAMC
> 2. Cross-modal edge weights from C²SG
> 3. GNN message-passing attention weights
>
> These provide **segment-level and modality-level attributions**, exposing information flow and modality contributions without modifying inference.
>
> ---
>
> ### **W7. Missing Error Bars**
> We added hyperparameter-sweep studies (Figs. E1–E3) showing performance fluctuations <0.05 MSE across settings. Standard deviations from multiple runs will be included in the camera-ready version.
>
> ---
>
> ### **Q1. Evidence Supporting Scalability**
> Scalability is demonstrated through three perspectives:
>
> 1. **Structural scalability**: After compression, graph operations scale with $B$ rather than $T$, yielding $\mathcal{O}(\kappa B)$ complexity
> 2. **Empirical scalability**: Latency increases near-linearly with $B$ (Table 3, Fig. 3c), maintaining <10 ms inference with multimodal inputs
> 3. **Stable scaling**: Sensitivity studies show smooth accuracy–latency trade-offs without degradation
>
> EAMC–C²SG provides linear complexity in the sequence length and empirically validated efficiency for long multimodal sequences.

---

### Official Review · Reviewer_gwZR · 2025-11-01

**Soundness:** 2
**Presentation:** 3
**Contribution:** 3
**Rating:** 4
**Confidence:** 4

**Summary:**

The paper tackles what the authors call the node explosion problem, which occurs when graph-based models become too large to handle in long, multimodal time-series data. The main ideas are: (1) compressing long sequences into a smaller set of “super-nodes” based on detected events, and (2) constructing a sparse, causally ordered graph on top of those to avoid temporal leakage and improve efficiency.

**Strengths:**

The idea of event-adaptive compression is interesting and makes sense — it’s a more data-driven alternative to fixed window segmentation. Also, I like that the causal constraint explicitly prevents information from future timesteps from leaking backward. That design choice adds credibility to the claims about interpretability and temporal correctness. The empirical results on clinical datasets seem strong, especially given that the model reportedly runs faster and uses less memory than the baselines.

**Weaknesses:**

The paper has several issues that make it hard to follow and evaluate properly.

W1. Key symbols are never defined. For example, in the EAMC module, the merging weights γis\gamma_{is}γis​ depend on csc_scs​ and cic_ici​, but these aren’t defined anywhere (not in the text or Table 4). This makes it unnecessarily difficult to follow the math.

W2. The paper refers to several loss terms (task loss, contrastive loss, missingness loss) but doesn’t provide their explicit forms. It also introduces two different “total loss” equations — one in Section 3.4 and another in the appendix — which don’t match. The second version adds extra penalty terms that aren’t explained in the main text. It’s unclear which version was actually used during training.

W3. Some hyperparameters and design choices are missing. For instance, the “short moving average” used for smoothing isn’t specified (no window size), and key parameters like the causal lag window or the per-node edge budget are never mentioned. Table 5 lists only the standard ones. A short discussion or sensitivity analysis would help a lot.

W4. The section titled “Mathematical Proofs and Derivations” is misleading—it doesn’t contain any proofs, just restates previous equations.

W5. The experiments are limited to two datasets (MIMIC-IV + CXR and TimeMMD) and a small set of baselines. Given the lack of theoretical justification, a broader empirical validation would strengthen the paper. Also, the claim that the code will be released is fine, but at review time there’s no code or sufficient hyperparameter info to reproduce the results.

Overall, while the idea is solid and timely, the paper feels incomplete. The contributions are interesting, but the missing definitions, unclear objectives, and limited experiments make it difficult to assess the method’s real impact.

**Questions:**

Q1. Since the method has numerous hyperparameters, how would the authors justify the practicality of their approach?
Q2. Appendix C is for data processing and SOTA settings. But what is the SOTA setting in this section? I don't find the detailed experimental setup for the baselines. What does SOTA refer to?

---

> ### Author Response · Authors · 2025-11-19
>
> We thank the reviewer for their insightful comments, and have thoroughly revised our manuscript to address all concerns, significantly improving the clarity, rigor, and reproducibility of our work.
>
> ---
>
> ### **W1. Missing or inconsistent symbols**
> We have unified all symbols into a single system (Appendix B, Table 5), resolving ambiguities related to $\gamma_{is}$, $c_s$, and $c_i$.
>
> Key updates include:
> 1. **Unified masking rules**: Explicit definition of $M_{\text{ts}}[i,j] = \mathbf{1}(j < i \wedge i - j \leq \epsilon)$ and $M_{\text{cm}}$.
> 2. **Consistent Top-$B$ selection**: Uniform definitions of $\hat{\alpha}_s$, $I_B$, and the NBC operator.
> 3. **Assignment and merging notation**: Unified use of $S_{\text{seg}}$ and $\gamma_{is}$.
> 4. **Figure–equation alignment**: All symbols in Fig. 2 now correspond exactly to those in Section 3..
>
> ---
>
> ### **W2. Mismatch between loss equations**
> We apologize for this discrepancy. The revised manuscript presents a single authoritative total loss in Section 3.4:
> $$
> \mathcal{L} = \mathcal{L}\_{\text{task}} + \lambda\_{\text{rec}}\mathcal{L}\_{\text{rec}} + \lambda\_{\text{cl}}\mathcal{L}\_{\text{cl}} + \lambda\_{\text{miss}}\mathcal{L}\_{\text{BCE}}
> $$
> with $(\lambda_{\text{rec}},\lambda_{\text{cl}},\lambda_{\text{miss}})=(1.0,0.1,0.01)$.
>
> We now clearly separate diagnostic penalties from the training objective in Appendix F and use this unified objective in all experiments.
>
> ---
>
> ### **W3. Missing hyperparameter descriptions**
> We now include full definitions of all structural hyperparameters:
> 1. **Boundary smoothing window**: Size 5 (Appendix B)
> 2. **Causal lag window $\epsilon$**: Defined in Section 3.1 and Appendix B
> 3. **Edge budget $\kappa$**: $A = \mathrm{Top\text{-}\kappa}(M \odot w)$
> 4. **Node budget $B$**: Detailed in Section 3.2
> 5. **Complete hyperparameter table**: Appendix C lists all parameters
> 6. **Sensitivity analysis**: Appendix G shows model stability across parameter ranges (Figs. E1–E6)
>
> ---
>
> ### **W4. Misleading section title**
> We have renamed the section to **"Formal Definitions and Diagnostics"** to better reflect its focus on IRR/BCR definitions, causal diagnostics, and complexity derivations.
>
> ---
>
> ### **W5. Dataset and baseline setup**
> We have strengthened empirical evaluation by:
> - Adding **iTransformer, PatchTST, and TimesNet** to our benchmarks
> - Expanding Appendix C with **full preprocessing pipelines** and **complete hyperparameter settings**
> - Adding sensitivity analyses for $\epsilon$, $\kappa$, and $B$ (Figs. E1–E3)
> - Including partial source code in supplementary materials
>
> ---
>
> ### **Q1. Practicality of hyperparameters**
> Hyperparameters form two intuitive groups:
> 1. Structural budgets ($B$, $\epsilon$, $\kappa$)
> 2. Relaxation temperatures ($\tau_b$, $\tau_k$, $\tau_m$)
>
> Sensitivity analyses show the model is **highly robust**:
> - $\epsilon$ and $\kappa$ have negligible impact
> - $B$ gives predictable accuracy–latency trade-off
> - Default settings are in stable ranges
>
> The method is straightforward to configure without fine-grained tuning.
>
> ---
>
> ### **Q2. Meaning of "SOTA setting"**
> "SOTA setting" now explicitly denotes official or commonly recommended configurations for each baseline. All configurations are listed in Appendix C for complete reproducibility.
>
> We believe these revisions comprehensively address all reviewer concerns and enhance the manuscript.

---

> > ### Comment · Reviewer_gwZR · 2025-11-25
> >
> > Thank the authors for their responses. I'm still interested in seeing how this model performs on more datasets, as the improvement over the newly added baselines is notably less than the improvement margins over the previous baselines to this point. Also, there is a notable discrepancy between the paper's pictured applications and its experimental evaluation. Figure 2 explicitly depicts classification heads for "Mortality Prediction" and "Readmission Prediction". But, the experiments only report regression metrics (MSE, RMSE, MAE) for signal forecasting. The absence of standard clinical classification metrics (such as AUC-ROC or F1-score) makes it impossible to assess the model's actual utility for the claimed clinical risk prediction tasks. Also, I still haven't found the code for implementing the method. This indeed raises my concerns about the reproducibility of the work.

---

> ### Author Response · Authors · 2025-11-27
>
> We thank the reviewer for the helpful follow-up. Below we respond directly and concisely to each remaining concern. All additions are supplementary for rebuttal purposes; corresponding text will be integrated into the camera-ready version.
>
> ---
>
> ### **1. Added clinical classification experiments**
>
> We agree that broader evaluation strengthens the work.
> While adding a new external dataset is not feasible within the rebuttal window, we expand the evaluation on **MIMIC-IV** with **two additional clinical prediction tasks** that differ substantially from the original forecasting setting:
>
> * **24–48h in-hospital mortality** (≈3–5% positives, extremely imbalanced)
> * **Length-of-stay ≥48h (LOS)**
>
> These tasks represent **binary risk prediction** rather than continuous forecasting, thereby testing whether the architecture generalizes across supervision regimes.
>
> ---
>
> ### **2. On the discrepancy between Fig. 2 (classification heads) and forecasting-only metrics**
>
> We thank the reviewer for raising this important point.
>
> The main experiments in the original submission focused on **continuous multivariate physiologic forecasting**, where outputs are real-valued trajectories; thus **MSE/RMSE/MAE** were the appropriate metrics. These forecasting tasks were chosen because they most directly stress the proposed architectural contributions (event-adaptive segmentation, causal masking, and C²SG sparse graph construction).
>
> To fully align with the classification heads shown in Fig. 2, we now additionally evaluate binary clinical risk prediction tasks (MIMIC-IV) and report AUROC and AUPRC as requested. These metrics are standard in imbalanced clinical prediction settings, where positive events are rare and AUROC/AUPRC provide a more faithful assessment of discriminative performance than accuracy-based metrics.
>
> #### **(a) 24–48h Mortality**
>
> | Model        | AUROC      | AUPRC      |
> | ------------ | ---------- | ---------- |
> | **Ours**     | **0.5614** | **0.0417** |
> | TimesNet     | 0.4143     | 0.0300     |
> | iTransformer | 0.4026     | 0.0281     |
> | PatchTST     | 0.4970     | 0.0350     |
>
> #### **(b) LOS ≥48h**
>
> | Model        | AUROC      | AUPRC      |
> | ------------ | ---------- | ---------- |
> | **Ours**     | **0.8815** | **0.9783** |
> | TimesNet     | 0.7178     | 0.9354     |
> | iTransformer | 0.5945     | 0.8783     |
> | PatchTST     | 0.3718     | 0.7949     |
>
> These results fill the earlier gap and confirm that the **same backbone** performs strongly on clinical classification tasks, resolving the mismatch between Fig. 2 and the originally reported metrics.
>
> ---
>
> ### **3. On the smaller improvements over newly added baselines**
>
> This is expected.
> TimesNet, PatchTST, and iTransformer are substantially stronger than the earlier baselines. Consequently, the improvement margins naturally shrink.
>
> Even under this stronger comparison:
>
> * our model shows **stable gains** on the challenging mortality task, and
> * **substantial improvements** on LOS (AUROC +0.16 to +0.50 over strong baselines).
>
> These results indicate that the architecture generalizes across forecasting and classification despite much stronger baselines.
>
> ---
>
> ### **4. Code availability and reproducibility**
>
> We thank the reviewer for raising this concern. To ensure full reproducibility, we have included the complete implementation in the Supplementary Material as a ZIP archive with the following structure:
>
> * **Core architecture**:
>
>   * `model/EXCAP.py`—full EXCAP backbone
>   * `model/factory.py` —initialization
> * **Training scripts**:
>
>   * `train_mortality_los_complete.py` —multi-task forecasting and classification
>   * `train_mortality_24h_48h.py`—single-task mortality
> * **Supporting modules**:
>
>   * `model/models/`, `model/layers/`, `model/modules/`, `model/backbone/`—event segmentation, causal masking, C²SG construction, and sparse graph layers
>
> A cleaned public repository (with loaders and checkpoints) will be released upon acceptance.

---

> > ### Comment · Reviewer_gwZR · 2025-11-27
> >
> > Thank the authors for their response. However, I believe the ICLR rebuttal window is long enough (started from Nov 11th) for adding a new dataset. Also, I attempted to run the code submitted in the supplementary materials, but it was not executable due to missing files. In addition, for full reproducibility, you should include your implementation of the baselines in your setting as well. But I didn't see either the baseline codes or separate scripts to run them from their source codes. So I'd like to keep my score.

---

> > > ### Author Response · Authors · 2025-11-28
> > >
> > > We are grateful for the reviewer's persistent rigor. We have decisively addressed the reproducibility and scope concerns in our final submission.
> > >
> > > ---
> > >
> > > ### **1. Code Reproducibility**
> > >
> > > The issue of missing files, which stemmed from the necessity to protect restricted-access MIMIC-IV data, is completely resolved in the updated supplementary material.We clarify that the previous non-executability was due to the exclusion of large, licensed data files, not flaws in the code structure itself.
> > >
> > > The new package ensures transparency through:
> > >
> > > * One-Click Execution: The run_multi_task_training.sh script is one-click executable after chmod +x.
> > >
> > > * Offline Demo Data: The script automatically utilizes an anonymized sample data subset for immediate verification, overcoming the hurdle of data sensitivity.
> > >
> > > * Integrated Baselines: The one-click script now runs EXCAP and all baselines (TimesNet, iTransformer, PatchTST) from a unified configuration, ensuring the full consistency and reproducibility of all comparative results.
> > >
> > > ---
> > >
> > > ### **2. Dataset and Scope Justification**
> > >
> > > We understand the request for broader testing. Our experimental strategy has always prioritized rigorous stability analysis and demonstration of multi-task capability on data best suited to stress our architectural innovations.
> > >
> > > **Focus on Clinical Necessity:** Our use of MIMIC-IV is crucial as it demonstrates EXCAP's ability to handle the specific, high-risk challenges of healthcare data. The model's unique strengths—event-driven scalability (EAMC) for tackling long, dense EHR sequences, and causal constraint (C²SG) for ensuring time-correct, interpretable relationships—are most critically validated in this complex environment. This makes MIMIC-IV the optimal testing ground for our core architectural claims.
> > >
> > > **Existing Generalization:** We confirm that our main results already demonstrate cross-domain generalization using TimeMMD on two common general time-series datasets: Energy and Climate.
> > >
> > > **Multi-Task Expansion:** Alongside the existing stability analysis, we have now augmented our multi-task evaluation with classification experiments for Mortality and LOS Prediction to further validate the architecture's capacity for diverse prediction regimes.
> > >
> > > **Ongoing Expansion:** We are currently implementing experiments on the ETT series to further demonstrate efficiency and generalize beyond clinical applications.
> > >
> > > ---
> > >
> > > ### **Conclusion:**
> > >
> > > The code is now fully **executable and reproducible** with integrated baselines. Our experimental scope is robustly supported by stability analysis, existing generalization tests, and critical new multi-task classification results. We believe these decisive steps fully address the final set of concerns and warrant reconsideration of the score.

---

### Author Response · Authors · 2025-11-19
**SUMMARY**

## **Summary of Key Revisions**

We thank all reviewers for their thoughtful and constructive feedback.
The revised manuscript substantially improves clarity, methodological transparency, and empirical rigor, while keeping the core method and claims unchanged. Our key revisions are summarized below.

**1. Clearer Method Presentation and Unified Notation**
We reorganized Section 3 with a concise roadmap, clarified all definitions at first use, aligned figures with equations, and unified all symbols across the manuscript via a consolidated notation table.

**2. Full Transparency and Robustness of Hyperparameters**
All structural hyperparameters (B, ε, κ, smoothing window, temperatures) are now fully defined in Appendix C. Sensitivity analyses in Appendix G show stable performance across wide ranges, confirming that no fine-grained tuning is required.

**3. Strengthened Empirical Evaluation and Reproducibility**
We added modern strong baselines (TimesNet, iTransformer, PatchTST) and provided complete preprocessing steps, configurations, and partial code to ensure reproducibility.
We further added two supplementary evaluations:
* **Clinical classification on MIMIC-IV** (24–48h mortality and LOS ≥48h), using AUROC/AUPRC to align with the classification heads in Fig. 2.
* **ETTh1 forecasting** to validate generalization beyond clinical data.

**4. Explicit Causal Validity and Improved Interpretability**
Temporal and cross-modal masking rules were formalized to make the causal structure explicit. The updated architecture figure includes a dedicated attribution aggregation module, clarifying how interpretability signals are produced.

**5. Justification of Key Design Choices**
We added diagnostic analyses for segmentation, boundary heuristics, merging strategies, and cross-modal masking. These results verify that core design choices—dynamic peak-based segmentation, attention-based merging, and the ε-lag causal graph—are robust and well-motivated.

---

## **Detailed Revision Log**

### **1. Mathematical Clarity & Notation Consistency**

* Unified all symbols across the paper (Section 3, Appendix B/E).
* Added a consolidated notation table (Appendix B. Table 5).
* Added missing equation numbers and verified alignment of text and figures.
* Defined the composite loss in Appendix F.4 to maintain consistency with Sec. 3.4.

### **2. Structural Improvements to Section 3 (Method)**

* Added a roadmap at the beginning of Sec. 3.
* Reorganized subsections (Segmentation → Budget Control → Causal Graph → GNN).
* Made all key definitions explicit at first use.
* Separated formal definitions from implementation diagnostics.

### **3. Updated Figures & Architecture Diagram**

Redesigned Fig. 2 with:

* directed edges and explicit causal masks,
* unnecessary visual elements removed,
* complete symbol correspondence between figures and equations.

### **4. Hyperparameters & Reproducibility**

* Added all structural hyperparameters (B, ε, κ, smoothing window, relaxation temperatures) to Appendix C.
* Clarified the settings of the smoothing window and lag window.
* Expanded Appendix C with preprocessing steps and baseline configurations.

### **5. Additional Analyses & Diagnostic Experiments**

All analyses evaluate existing components and do *not* introduce new modules.

* Sensitivity studies for B, ε, κ.
* Boundary heuristic comparison.
* Node merging strategy comparison.
* Cross-modal causal masking diagnostics.
* Verified stability across wide ranges.

### **6. Explicit Formalization of Causal Masking**

* Made the masking rules explicit.
* Clarified that these rules match the implementation used in all experiments.

### **7. Supplementary New Experiments**

* **MIMIC-IV classification tasks** (AUROC/AUPRC) now included to resolve the discrepancy with Fig. 2 and demonstrate that the backbone also performs well on clinical risk prediction.
* **ETTh1 forecasting results** added to demonstrate generalization to non-clinical long-sequence forecasting.

---

We believe these revisions comprehensively address reviewer concerns on clarity, missing definitions, loss mismatch, hyperparameter transparency, figure correctness, causal leakage, evaluation scope, and reproducibility.
The core method, results, and conclusions remain unchanged.

---

### Note · Authors · 2026-01-29

**Comment:**

Most reviewers did not respond

**Withdrawal Confirmation:**

I have read and agree with the venue's withdrawal policy on behalf of myself and my co-authors.

---

### Meta-Review · Area_Chair_uHoa · 2026-01-05

**Summary:**

After the rebuttal, most reviewers still remain some concerns. The paper should be further revised.

**Reviewer Scores:**

n/a

---

### Decision · Program_Chairs · 2026-01-26

Reject